



# Clouds over Hyytiälä, Finland: an algorithm to classify clouds based on solar radiation and cloud base height measurements

Ilona Ylivinkka[1,2], Santeri Kaupinmäki[1,*], Meri Virman[1], Maija Peltola[1], Ditte Taipale[1,2],
Tuukka Petäjä[1], Veli-Matti Kerminen[1], Markku Kulmala[1], and Ekaterina Ezhova[1]

[1]Institute for Atmospheric and Earth System Research / Physics, Faculty of Science, University of Helsinki, P.O. Box 64,
00014 Helsinki, Finland
[2]SMEAR II station, University of Helsinki, 35500 Korkeakoski, Finland
*_Current address:_ Department of Medical Physics and Biomedical Engineering, University College London, London, WC1E
6BT, United Kingdom

**Correspondence:** Ilona Ylivinkka (ilona.ylivinkka@helsinki.fi)

**Abstract.** We developed a simple algorithm to classify clouds based on global radiation and cloud base height measured by pyranometer and ceilometer, respectively. We separated clouds into seven different classes (stratus, stratocumulus, cumulus, nimbostratus, altocumulus+altostratus, cirrus+cirrocumulus+cirrostratus and clear sky+cirrus). We also included classes for cumulus and cirrus clouds causing global radiation enhancement, and classified multilayered clouds, when captured by the

ceilometer, based on their height and characteristics (transparency, patchiness and uniformity). The overall performance of the algorithm was nearly 70 % when compared with classification by an observer using total sky images. The performance was best for clouds having well-distinguishable effects on solar radiation: nimbostratus clouds were classified correctly in 100 % of the cases. The worst performance corresponds to cirriform clouds (50 %). Although the overall performance of the algorithm was good, it is likely to miss the occurrence of high and multilayered clouds. This is due to the technical limits of

the instrumentation: the vertical detection range of the ceilometer and occultation of the laser pulse by the lowest cloud layer.

    We examined the use of brightness parameter, which is defined as a ratio between measured global radiation and modeled radiation at the top of the atmosphere, as an indicator of clear sky conditions. Our results show that cumulus, altocumulus, altostratus and cirriform clouds can be present when the parameter indicates clear sky conditions. Those conditions have previously been associated with enhanced aerosol formation under clear sky. This is an important finding especially in case

of low clouds coupled to the surface which can influence aerosol population via aerosol-cloud interactions. Overall, caution is required when the parameter is used in the analysis of processes affected by partitioning of radiation by clouds.

## 1   Introduction

Clouds regulate the radiative heating of the Earth because they reflect a large share of the incoming solar radiation back to space, and also absorb and re-emit long-wave radiation radiated by the Earth (Schneider and Dennett, 1975; IPCC, 2013). The

light scattering and absorption properties of clouds depend on their thickness and spatial distribution, but also on the size and phase of cloud droplets. These characteristics, in turn, vary for different types of clouds. For example, optically thick stratiform




clouds effectively decrease the amount of solar radiation reaching the surface of the Earth, thereby cooling the climate. The dominant impact of optically thin and transparent cirrus clouds is mainly on the outgoing long-wave radiation, leading to a net warming effect (IPCC, 2013). Additionally, clouds alter the ratio between direct and diffuse radiation on the surface of the

Earth (Kasten and Czeplak, 1980; Calbó et al., 2001). Hence, the cloud properties largely affect the radiation budget of the Earth (Sinha and Shine, 1995; Loeb et al., 2009) as well as many physical and chemical processes in the planetary boundary layer (Gu et al., 1999; Mogensen et al., 2015; Jokinen et al., 2017). Many of the cloud-related interactions and feedbacks are not well understood, causing large uncertainties in the predictions of the future climate change (IPCC, 2013).

Shortwave global radiation comprises direct radiation coming from the direction of the Sun, and diffuse radiation coming

from all other directions due to scattering of solar radiation in the atmosphere. Under clear sky conditions, 10–20 % of global radiation is diffuse radiation, depending on the aerosol load in the atmosphere and time of the day. When clouds overcast the sky, diffuse radiation is nearly equal to global radiation (Page, 2012). Additional effect related to partitioning of solar radiation by clouds is global radiation enhancement (GRE), which means that the measured global radiation exceeds the theoretical maximum clear sky radiation, and is associated with specific "focusing" of radiation by clouds (Pecenak et al., 2016).

Partitioning of radiation by clouds affects on ecosystem and atmospheric processes. For example, under diffuse radiation conditions, the photosynthesis of a forest ecosystem is more effective. Such enhancement is presumably caused by the facts that diffuse radiation penetrates more evenly inside the canopy so that more leaves can photosynthesize efficiently, and that photosynthetic saturation of the leaves on top of the canopy is less likely to be reached (Gu et al., 2002; Kivalov, 2018). In cloudy conditions, the increase in gross primary production, which is a measure of ecosystem-scale photosynthesis, can be up

to 30 % compared to clear sky and clean atmosphere conditions in boreal forests (Ezhova et al., 2018).

The presence of clouds modulates also atmospheric chemistry. For example, the production of OH, which is the most important oxidant in the atmosphere, is reduced when clouds limit the incoming ultraviolet radiation, thereby reducing also the oxidation of e.g. biogenic volatile organic compounds (BVOC) (Atkinson and Arey, 2003; Mogensen et al., 2011, 2015; Hellén et al., 2018). Oxidized BVOCs form vapors that are able to contribute to the formation and growth of atmospheric

aerosol particles (Hallquist et al., 2009; Riipinen et al., 2012; Donahue et al., 2013; Schobesberger et al., 2013; Ehn et al., 2014; Kulmala et al., 2014b; Riccobono et al., 2014). The changes in aerosol processes additionally affect cloud condensation nuclei (CCN) production in the atmosphere (Kerminen et al., 2012; Paasonen et al., 2013; Scott et al., 2018; Sporre et al., 2019), altering also several cloud properties, such as their albedo and lifetime, their ability to precipitate, and cloudiness in a more general sense (Twomey, 1977; Albrecht, 1989; Gryspeerdt et al., 2014; Rosenfeld et al., 2014). Hence, the modulations

in cloudiness affect variety of different simultaneous processes and feedbacks, and the research on these interactions requires accurate knowledge of different cloud types and their effects on radiation on different time scales (Hussein et al., 2009; Rannik et al., 2013; Dada et al., 2018; Ezhova et al., 2018).

Measurements at SMEAR II (Station for Measuring Ecosystem–Atmosphere Relations) in Hyytiälä, Finland aim for comprehensive understanding of the ongoing processes in the atmosphere, ecosystem and the interactions between them (Hari and

Kulmala, 2005). Despite the importance of the clouds on these processes, to date the prevailing cloud types have not been iden-





tified. The objective of this work is to formulate a simple and inexpensive algorithm to estimate the cloud types over SMEAR II based on its existing instruments.

Traditionally, cloud type classification has been based on human observations. However, human observations are not always available, especially in remote locations, and the time resolution of the data is too low for many scientific applications. Thus, automated cloud classification methods have been developed. Either data from ground or satellite instrumentation can be used for the classification. The equipment for this purpose include cameras, radiosondes, and different kinds of irradiance meters and radars (Tapakis and Charalambides, 2013). The classification applied in simple models can base only on one instrument (Duchon and O'Malley, 1999), or the algorithm can employ data from several instruments (Wang and Sassen, 2001). Ground-based measurements provide accurate results on local variations in cloudiness, whereas satellite measurements cover large-scale phenomena (Duchon and O'Malley, 1999; Ricciardelli et al., 2008).

In instrumental-based cloud classification, the clouds can be classified according to e.g. the attenuation of irradiance compared to theoretical clear sky values, or meteorological variables, such as temperature thresholds. Image-based classification employs spectral and textural features of an image. For example, the tonal variation may help in distinguishing between different types of clouds (e.g. cirrus and cumulus), and the spatial homogeneity allows to discriminate between similar types of clouds (e.g. cumulus and stratocumulus) (Haralick et al., 1973; Calbó and Sabburg, 2008; Heinle et al., 2010). The algorithms calculating the cloud occurrence can be very simple separating only clouds from the background (Cayula and Cornillon, 1996; Long and Ackerman, 2000; Mukherjee and Acton, 2002), or sophisticated classifying different cloud types into several classes (Calbó et al., 2001; Bankert and Wade, 2007; Ricciardelli et al., 2008). The classification can be based on exceeding linear threshold values (Kegelmeyer Jr, 1994), or it can apply machine learning and artificial intelligence with large training sets (Bankert and Wade, 2007; Mazzoni et al., 2007).

The selection of a suitable method depends on the application of the results. For example, Cloudnet measurement stations, producing cloudiness data for the needs of weather forecasting, have at least three instruments providing information of cloud vertical structure, and ice and liquid water contents (Illingworth et al., 2007). The main instruments include dopplerized cloud radar, ceilometer and dual-frequency microwave radiometer. The calibration and data handling processes are exact and pre-arranged (Illingworth et al., 2007). While overall, Cloudnet provides very detailed information of clouds, for some applications this information is redundant. As an example, when dealing with the processes related to solar radiation, it is reasonable to characterize clouds using solar radiation as a classification parameter. Moreover, Cloudnet stations and the instruments they use are rare, while for example global radiation and cloud base height (CBH) are often measured routinely.

Here, we introduce an automatic method to classify clouds based on global radiation and CBH measurements. Our algorithm is an adaptation of the work by Duchon and O'Malley (1999). Their so called "pyranometer method", using only pyranometer data, was developed to classify clouds in places where no human observations were available (Duchon and O'Malley, 1999). Even though the pyranometer method is simple and effective, its cloud type classes are rather broad (stratus, cumulus, cumulus+cirrus, cirrus, clear sky, precipitation+fog, and other), and the classification was found to be in agreement with human observations only 45 % of the time (Duchon and O'Malley, 1999). Our improved cloud type classification algorithm uses additionally CBH data. Hence, the number of cloud type classes can be increased compared to Duchon and O'Malley (1999)



because the clouds at different levels can be distinguished. Cloud classes in our algorithm are cumulus, stratus, strotocumulus, nimbostratus, altocumulus+altostratus, cirrus+cirrocumulus+cirrostratus, clear+cirrus, cumulus+GRE, and Ci+GRE. Although the algorithm is developed using the data from one measurement station, it can be applied also to other environments.

In order to illustrate the application of the new cloud classification algorithm we study the cloud statistics over Hyytiälä. In
the future, the results of this algorithm may be employed in other analyses regarding cloud-related interactions and feedbacks. This is possible due to the fact that the data set including ceilometer and pyranometer data from SMEAR II is ten years long, compared to just few years' data set of more advanced cloudiness measurements (e.g. Cloudnet), and one year of total sky imagery from Hyytiälä.

## 2    Materials and methods

We develop a cloud classification algorithm, utilizing global radiation and CBH data, to identify cloud types and analyze the statistics pertaining to cloudiness. In Sect. 2.1 we first introduce measurement site, instruments and data set. The radiation based parameters employed for the cloud classification are derived in Sect. 2.2, and in Sect. 2.3 we describe how cloud occurrence can be estimated using only pyranometer data.

### 2.1    Site and data set

SMEAR II in Hyytiälä in southern Finland (61°51′N, 24°17′E, 180 m a.s.l.) is a background measurement site. The state of the atmosphere and ecosystem are monitored with various instruments to understand the ongoing processes, interactions and feedbacks. The station, surrounded by 57-years-old Scots pine (*Pinus sylvestris*) dominated forest, was established in 1995 (Hari and Kulmala, 2005).

The main data set in this work includes data from a pyranometer and a ceilometer. The pyranometer (Middleton solar SK08
pyranometer) measures global radiation at wavelengths of 0.3–4.8 μm. The ceilometer (Vaisala CL31) detects CBHs for a maximum of three different cloud layers based on the back-scattering profile of a laser pulse. Its maximum measurement height is 7 500 m. Data points with full and partial obscuration, occurring usually during rain or fog events, have been excluded from the analysis.

The measured global radiation ($I_{\mathrm{meas}}$) is compared to modeled clear sky radiation ($I_{\mathrm{gh}}$) to quantify how effectively clouds
block radiation. To calculate the modeled clear sky radiation, we used Solis clear sky model (Ineichen, 2008). The model is different from that of Duchon and O'Malley (1999), in which precipitable water was estimated based on dew point and aerosol optical depth (AOD) was taken constant. We used Solis model because it explicitly takes into account the aerosol load in the atmosphere. The input parameters are measured AOD at 700 nm and precipitable water. We used AOD at 675 nm and precipitable water obtained from Aerosol Robotic Network (AERONET) data base for Hyytiälä (Holben et al., 1998). Note,
however, that the data from 2014 is found under the name "ARM Hyytiälä Finland" because in 2014 Atmospheric Radiation Measurement (ARM) facility of the U.S. Department of Energy had a campaign called "Biogenic Aerosols – Effects on Cloud



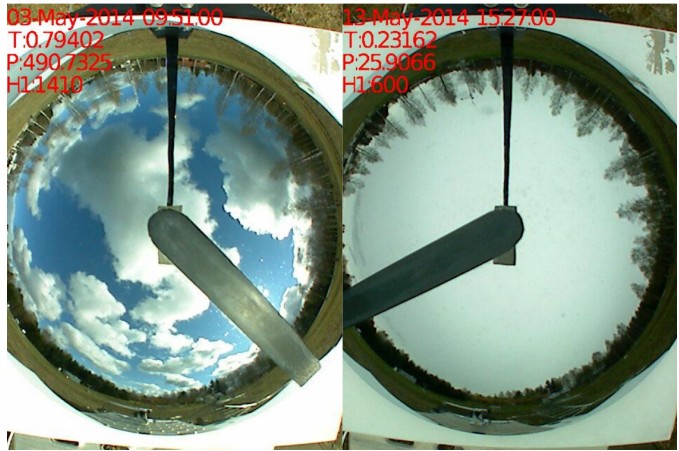

**Figure 1.** An example of total sky images taken in Hyytiälä that were used when formulating the algorithm. Transparency, patchiness and the lowest CBH are marked in the figures. Figure courtesy of ARM.

and Climate" (BAECC) in Hyytiälä (Petäjä et al., 2016). We used version 2 and level 2 (cloud screened and quality controlled) AERONET data. The data are available at https://aeronet.gsfc.nasa.gov/ (last access: 04 February 2020).

In order to reduce the overlap of parameter ranges between different cloud types (Duchon and O'Malley, 1999), and to make

sure that the parameter ranges are applicable for conditions in Hyytiälä, we compared the cloud classification made by a human observer, using total sky images, to corresponding radiation characteristics and CBHs between 01 May and 31 July 2014 (Fig. 1). In the validation process, we used total sky images and ceilometer data from the BAECC campaign. The ceilometer used in the campaign was also Vaisala CL31, but it was positioned about 500 m away from the standard ceilometer of SMEAR II. We discuss the consequences of ceilometer position for cloud classification in Appendix A1 (Fig. A1).

In the cloudiness and cloud classification analysis, we used quality checked pyranometer and ceilometer data measured at SMEAR II in 2014 and 2016–2017. Data from 2015 was excluded because the data availability was low due to instrumental issues of the ceilometer. The time resolution of the data was 1 min, and gaps were interpolated with the nearest value. The interpolation was important only for intermittent measurements of precipitable water and AOD. Otherwise the data availabilities of the measured variables were high during the measurement period, ca. 90 %. When conducting seasonal analysis, we

determined the seasons so that spring included March, April and May, followed by summer (June, July and August), autumn (September, October and November) and winter included (December, January and February).

For the cloud type classification and cloud occurrence analysis based on pyranometer measurements, we used only data when solar zenith angle (SZA) was less than 70° as the pyranometer data are not reliable when the Sun is close to horizon. Because SZA is always larger than 70° before 27 February and after 16 October, we included only data from March to

September so that we used only months with full data availability. However, for the cloud occurrence and CBH analysis using the ceilometer measurements, we used data independent of the time of day and season, because the ceilometer is not as sensitive





to SZA as the pyranometer. We calculated the value of SZA with Solar Position Algorithm (SPA) online calculator, available in https://midcdmz.nrel.gov/solpos/spa.html (last access: 09 January 2020).

## 2.2 Cloud type classification parameters

Our algorithm uses three parameters to classify clouds: transparency (TR), patchiness (PA) and measured CBH. Transparency is the ratio of the measured global radiation ($I_{meas}$) to the modeled clear sky radiation ($I_{gh}$) averaged over a running time interval:

$$\text{TR} = \left\langle \frac{\text{measured global radiation}}{\text{modeled clear sky radiation}} \right\rangle_{21\ \text{min}} = \left\langle \frac{I_{meas}}{I_{gh}} \right\rangle_{21\ \text{min}}. \tag{1}$$

Transparency describes how effectively clouds block solar radiation. Transparency is equal to 1 in clear sky conditions and
it approaches 0 for an overcast sky. In this work the chosen time interval is 21 min similar to Duchon and O'Malley (1999). The length of the time interval is based on empirical experience: the time interval should be long enough to capture the cloud variability, but it should not be too long so that the prevailing cloud type changed within one interval.

Patchiness is the running standard deviation ($\sigma$) of scaled measured global radiation ($I_{sc,meas}$):

$$\text{PA} = \sigma \left( \frac{\text{measured global radiation} \times 1400\ \text{Wm}^{-2}}{\text{modeled clear sky radiation}} \right)_{21\ \text{min}} = \sigma \left( I_{sc,meas} \right)_{21\ \text{min}}. \tag{2}$$

Patchiness determines the variability of the cloud layer. The same time window of 21 min is used. The scaling of the global radiation is discussed later in this section.

The third criterion is the running minimum of the lowest CBH over a 21 min interval. To assess the patchiness of cumulus clouds, an additional parameter ($\text{TR}_{max}$) is included. $\text{TR}_{max}$ is 21 min moving maximum of the relation between measured global radiation and modeled clear sky radiation, and hence it describes the cloud free moments when cumulus clouds are
present.

In this study, we used Solis clear sky model to calculate the amount of global radiation that would reach the surface of the Earth in case there were no clouds (Ineichen, 2008). From the model, the obtained global radiation at ground level is

$$I_{gh} = I_0' \cdot \exp \left( \frac{-\tau_g}{\cos^g(\text{SZA})} \right) \cdot \cos(\text{SZA}), \tag{3}$$

where $I_0'$ is the solar flux density at the top of the atmosphere ($I_0$) multiplied by a factor associated with AOD and precip-
itable water, $\tau_g$ is global total optical depth, and $g$ is a fitting parameter related to AOD and precipitable water. The detailed descriptions of the parameters can be found in Ineichen (2008).

The relationship between the measured global radiation and the modeled global radiation gives the fraction of radiation that reaches the surface of the Earth. For the cloud classification algorithm, we scaled the measured radiation because the magnitude of the oscillations in the radiation due to clouds are different depending on the time of day. Since the amount of incoming solar
radiation is lower in the morning and evening compared to the noon, the fluctuations due to same types of clouds are higher around noon. We used $1400\ \text{Wm}^{-2}$ for scaling because it is slightly higher than the theoretical maximum of incoming solar



radiation. The scaling factor ($s$) was calculated as in Duchon and O'Malley (1999):

$$s = \frac{1400 \text{ Wm}^{-2}}{I_{\text{gh}}}. \tag{4}$$

We multiplied the measured global radiation by the scaling factor in order to obtain the scaled radiation:

$\quad I_{\text{sc,meas}} = I_{\text{meas}} \cdot s. \tag{5}$

## 2.3 Calculating cloud occurrence from pyranometer data

We determined the cloud occurrence from the pyranometer measurements as the ratio between the measured global radiation and modeled radiation at the top of the atmosphere ($I$). The radiation at the top of the atmosphere is

$$I = \cos(\text{SZA}) \cdot I_0. \tag{6}$$

We used 21 min running average of data in 1 min time resolution. When the radiation measured with the pyranometer was less than a certain percentage of the modeled top of the atmosphere radiation, we assumed that the data point corresponded to cloudy conditions. For summer months, the percentage that we used was 70 %, for April 65 %, and for March and September 55 %. We estimated the percentages separately for each month using a clear sky model with relatively high aerosol load ($\text{AOD}_{675 \text{ nm}}$ = 0.17). The percentages were different for different months because the position of the Sun is higher in summer than in spring 185 and autumn. For this analysis, we used only data from March to September to avoid errors in the measurements caused by large SZA.

## 3 Results and discussion

First, we analyzed the ceilometer and the pyranometer data to study the seasonal variation in cloud occurrence in order to gain insight into how often clouds are observed over Hyytiälä, and what are the typical CBHs (Sect. 3.1). Second, the cloud 190 classification algorithm is introduced along with the evaluation of the performance of the algorithm (Sect. 3.2). We study the statistics of the automatically produced cloud types in Sect. 3.3. In Sect. 3.4, we discuss the use of brightness parameter as an indicator of clear sky condition, and finally in Sect. 3.5 we compare the main findings of this work with other studies.

### 3.1 Cloud properties

We studied the seasonal variation of cloud occurrence measured with the ceilometer (Fig. 2a) and the pyranometer (Fig. 195 2b). We obtained the cloud occurrence by dividing the number of cloud observations in one month by the total number of data points in the month. Cloud occurrence calculated using ceilometer data (Fig. 2a) included all the data, whereas cloud occurrence calculated using the pyranometer data (Fig 2b) included only times when SZA was less than $70°$. The ceilometer measurements show a robust seasonal variation in cloud occurrence in Hyytiälä, with cloud occurrence being lower during summer months (56 %) and higher during winter months (79 %) (Fig. 2a). The overall cloud occurrence was about 66 %, and





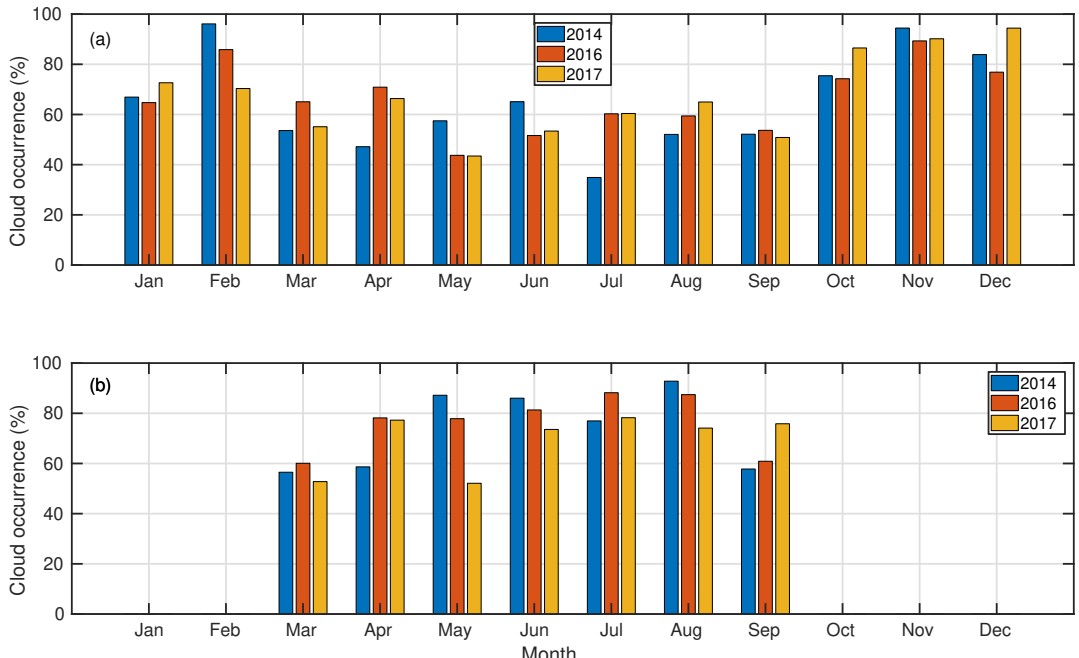

**Figure 2.** The monthly average cloud occurrence observed over Hyytiälä based on (a) the ceilometer data and (b) the pyranometer data. The pyranometer data were limited to values when SZA was less than $70°$.

from March to September it was about 55 % (Fig. 2a). Cloud occurrence calculated from the pyranometer data did not show a seasonal cycle, and also the cloud occurrence was higher (73 %) compared to the ceilometer measurements (Fig. 2b).

Diurnal variation of cloud observations by the ceilometer in different seasons is shown in Fig. B1. A diurnal cycle in cloud occurrence was observed in summertime (Fig. B1b). The cloud occurrence had a maximum around 14:00, likely being associated with the development of convective clouds. In May and September, a robust diurnal cycle was also observed whereas in other months the variation was absent (Fig. B1a and c). We did not investigate the diurnal variation from the pyranometer measurements as the method is limited by SZA, and hence the observations were not distributed evenly throughout the day.

From the ceilometer data, we could retrieve the occurrence of the two-layered and three-layered clouds. The second and the third cloud base were observed about 2–10 % and less than 1 % of the time, respectively, depending on the month (Fig. B2). Hence, the frequency of the times when single-layered clouds were detected by the ceilometer, was approximately the same as the observed cloudiness in total. Both the second and third cloud layer seemed to have higher frequencies of occurrence during summertime compared to winter, even though there were substantial differences between the years (Fig. B2). When a multilayered cloud was observed, it was two-layered in 92 % of the cases.

To identify the most common CBHs observed over Hyytiälä, we investigated the seasonal (Fig. 3) and diurnal (Fig. B3) variation of CBHs measured with the ceilometer. In each month, we divided every CBH record from the lowest cloud layer into 400 m bins. We calculated the frequency of CBH records in each bin as a ratio between the number of CBH records in





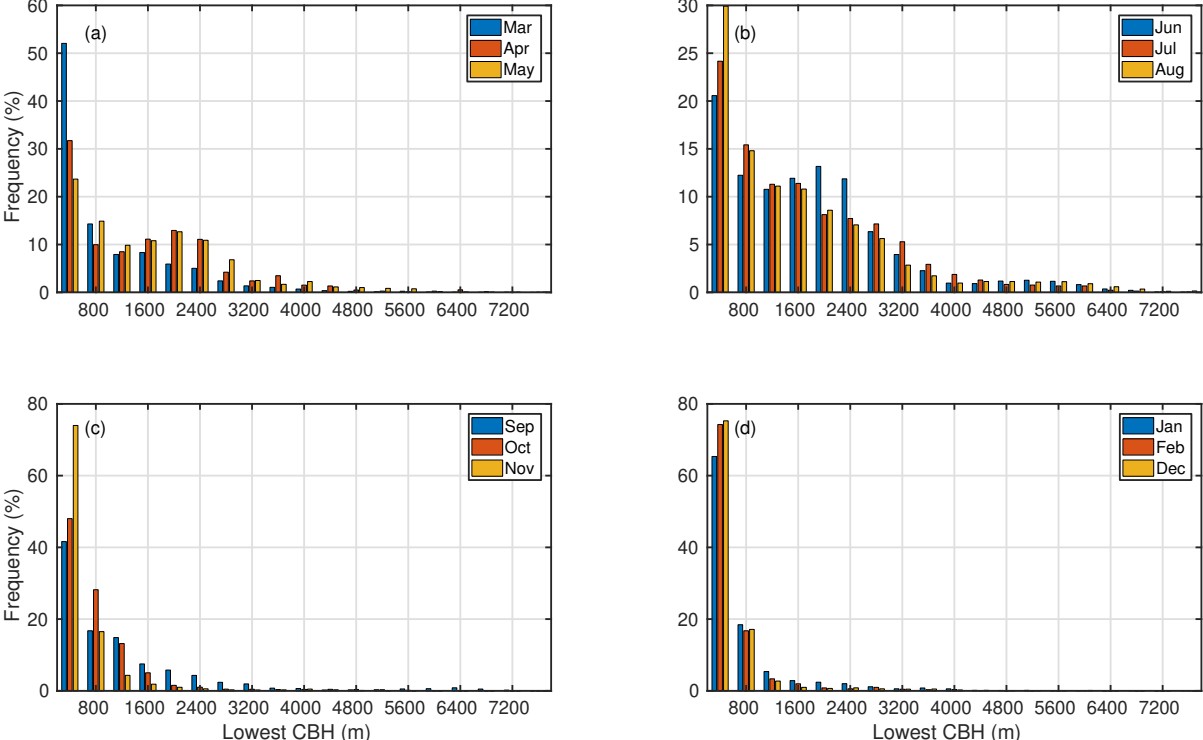

**Figure 3.** The number of CBH observations, from the lowest level of the ceilometer, in each 400 m height bin in one month with respect to the total number of observed cloud bases in that month in (a) spring, (b) summer, (c) autumn and (d) winter months. Notice the differences in y-axes scaling.

the bin and the total number of CBH records in that month. Figure 3 shows that when a cloud base was observed, the most frequently observed CBH was below 800 m for all months, although the relative amount of CBH records below 800 m was higher in winter than in summer. Figure 3 also shows that in spring and summer, the CBH distribution was more dispersed, and a second maximum at about 1 600 m was detected. The measured CBHs were most likely associated with cumulus clouds as they were more often observed in summer (see Fig. 5 in Sect. 3.3). This is also supported by the fact that a pronounced diurnal cycle in CBH, with higher values in afternoon compared to morning, was measured in summer, whereas in winter no diurnal cycle was observed (Fig. B3). In summertime, the frequently detected CBHs around 3 000–4 000 m are probably middle level altocumulus and altostratus clouds. Overall, low clouds (CBH < 2000 m) were observed 87 % of the time when clouds were detected, middle clouds (2000 m < CBH < 7000 m) 13 %, and high clouds (CBH > 5000 m) 1 % of the time. The limits of the cloud level classification follow the criteria defined in Houze (1993).

The seasonal variation of CBH distribution of all clouds, and single, two and three-layered clouds, are presented in Fig. B4. It confirms the observation of single-layered clouds dominating the CBH distribution based on ceilometer data that we found





also in Figs. 2 and B2, as the distribution of single-layered clouds resembles the distribution of total observed cloudiness. The seasonal variation of the CBHs of multilayered clouds reflects the seasonal variation of the lowest cloud layer (Fig. 3).

Figure B5 displays the height differences between cloud layers in different seasons. In all the seasons, the most common height difference between all the layers (1st and 2nd in two-layered clouds, 1st and 2nd in three-layered clouds and 2nd and 3rd in three-layered clouds) was less than 400 m (50–70 %). Towards the higher end, the distribution decreased gradually. The large distances between cloud layers were slightly more common in autumn and winter than in spring and summer (Fig. B5). In cases when three cloud layers were detected, the distance between the first and the second cloud layer was usually

smaller than the distance between the second and the third cloud layer. Also, the first and the second cloud layer, in cases with three-layered clouds, were found more often close to each other (distance less than 400 m) than the first and the second cloud layer in two-layered cloud cases (Fig. B5).

## 3.2    New cloud classification algorithm and its evaluation

The algorithm classifies clouds based on three parameters, determined in Sect. 2.2: CBH, transparency and patchiness. We

adjusted the ranges of the parameters, corresponding to different types of clouds, by constructing planes (TR,PA) for different types of clouds based on total sky images from Hyytiälä (Fig. 1). We took uniformly and randomly a sample of 665 total sky image–measurement data pairs. To ensure that the middle and high clouds were represented in the analysis, we took another sample of 320 pairs with the condition that the minimum CBH at that time was at least 2 000 m.

We first classified the clouds into cumulus, stratus, stratocumulus, nimbostratus, altocumulus, altostratus, cirrus, cirrocumu-

lus, cirrostratus or clear sky, and the corresponding transparency, patchiness and CBH were recorded. We put the transparency and patchiness values in the plane of parameters (TR,PA) in order to determine the regions in the plane, corresponding to different cloud types. Some cloud types had a significant overlapping in the plane of parameters (TR,PA), and thus could not be distinguished from each other from the point of view of their influence on solar radiation. Those we combined in more general cloud classes (altocumulus and altostratus, cirrus, cirrocumulus and cirrostratus, and clear sky and cirrus). The clear

sky class contains also cirrus clouds because cirrus clouds are difficult to distinguish from clear sky as they are transparent and may have non-detectable CBH. Hence, the final cloud classes used in this study are cumulus (Cu), stratus (St), stratocumulus (Sc), nimbostratus (Ns), altocumulus+altostratus (Ac+As), cirrus+cirrocumulus+cirrostratus (Ci+Cc+Cs) and clear+cirrus (clear+Ci). Additionally, we defined separate classes for cumulus and cirrus clouds that caused global radiation enhancement (Cu+GRE and Ci+GRE, respectively).

We created rectangular segmentations in the (TR,PA) plane based on those cloud classes, and thus gained the new parameter ranges for each cloud type. After that, we implemented the parameter ranges into the cloud type classification algorithm. The whiskers inserted into Fig. 4 indicate the transparency and patchiness ranges for different cloud type classes. The CBHs and parameter ranges of radiation characteristics for all cloud types are listed in Table 1.

If the parameters did not fit to the parameter ranges of any of the listed cloud types (Table 1), or the data were missing,

we classified the cases into separate classes based on whether the ceilometer did ("Base, no class") or did not ("No base, no class") capture a cloud base. As the ceilometer data were quality checked, the latter class contains basically data points when

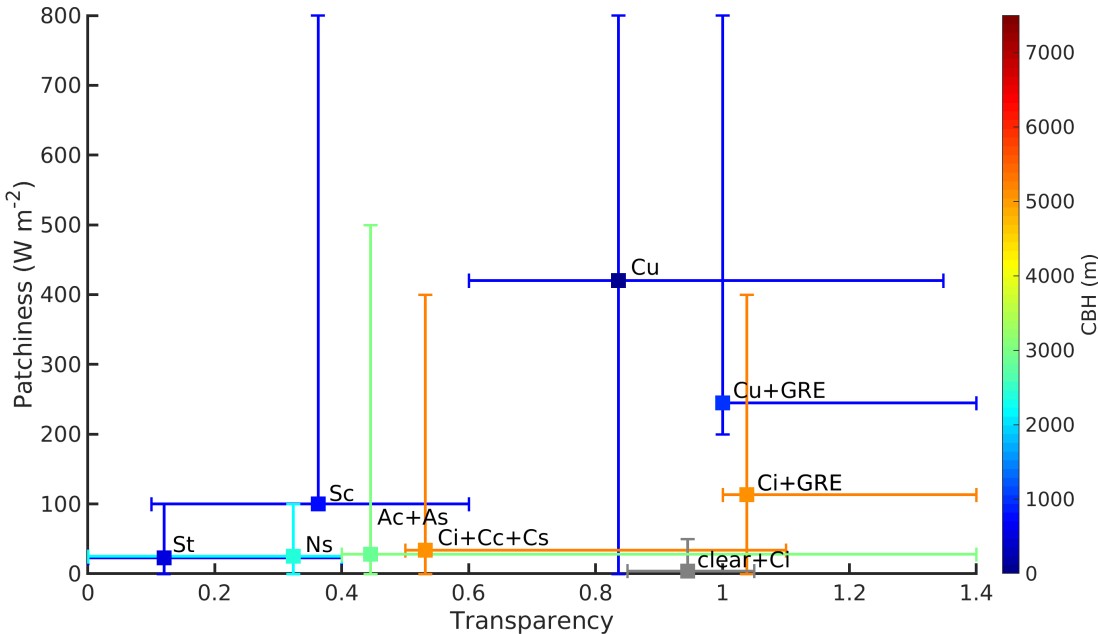

**Figure 4.** Transparency and patchiness ranges used as classification criteria for different cloud types (whiskers). Markers display the locations of the maximum data point density for each cloud type.

the sky was clear but the values of other parameters did not fit to the clear sky parameter ranges, or the data were missing. Additionally, as the field of view of the ceilometer is narrow, the class contains data points when the cloud was not in the field of view of the ceilometer although there were clouds present.

We also examined the characteristics of the second and third cloud layer in cases of multilayered clouds as measured by the ceilometer. We classified these clouds based on the height of the second or third cloud base, and characteristics defined by transparency and patchiness. We used three height classes: "Low level" (LL), "Middle level" (ML) and "High level" (HL). If the second or third cloud base was below 2000 m, the case was classified as "Low level", if between 2000 and 7000 m situation was "Middle level", and if above 5000 m the situation was "High level" (Houze, 1993). We classified the different

cloud layers separately. Hence, even though e.g. the second cloud layer was middle level cloud, simultaneously there might also exist low or high level cloud layers. As other cloud layers could be difficult to be detected above the first cloud layer with the ceilometer, additional condition low level cloud layer was determined: if the difference between the 21 min moving maximum and minimum CBH ($CBH_{max} - CBH_{min}$) of the lowest cloud layer was more than 1000 m, and $CBH_{max}$ was less than 2000 m, the case was considered as low level multilayered cloud (Table 1).

We divided multilayered clouds into three characteristic classes: "Multilayer uniform" (MuUni) for uniform and thick cloud layers such as stratus and nimbostratus, "Multilayer transparent" (MuTr) for uniform and transparent cloud layers like cirrostratus and "Multilayer patchy" (MuPa) for patchy clouds with varying transparency such as altocumulus. However, the actual





**Table 1.** The cloud types and corresponding parameter ranges used in the algorithm to determine the different cloud types. Nr of layer refers to the number of the cloud layer that is used as a criterion. Notice that cumulus, cirrus+cirrocumulus+cirrostratus and low level multilayered cloud classes have multiple criteria (see also Sect. 3.2)

| Cloud type | CBH (m) | Transparency | Patchiness (Wm$^{-2}$) | Nr of layer |
|---|---|---|---|---|
| Cumulus (Cu) | $< 2000$ | $0.6$–$0.85$ & $TR_{max} > 1$ | $\geq 200$ | 1 |
| | $< 2000$ | $> 0.85$ & $TR_{max} > 1$ | $> 0$ | 1 |
| Stratus (St) | $< 2000$ | $< 0.4$ | $< 100$ | 1 |
| Stratocumulus (Sc) | $< 2000$ | $0.1$–$0.6$ | $\geq 100$ | 1 |
| Nimbostratus (Ns) | $2000$–$3000$ | $< 0.4$ | $< 100$ | 1 |
| Altocumulus+Altostratus (Ac+As) | $2000$–$5000$ | $\geq 0.4$ | $< 500$ | 1 |
| Cirrus+Cirrocumulus+Cirrostratus | $\geq 4000$ | $0.85$–$1.1$ | $50$–$400$ | 1 |
| (Ci+Cc+Cs) | $\geq 4000$ | $0.5$–$0.85$ | $< 400$ | 1 |
| Clear+Cirrus (Clear+Ci) | NaN | $0.85$–$1.05$ | $< 50$ | 1 |
| Cumulus+GRE (Cu+GRE) | $< 2000$ | $> 1$ & $TR_{max} > 1$ | $\geq 200$ | 1 |
| Cirrus+GRE (Ci+GRE) | $\geq 4000$ | $> 1$ | $< 400$ | 1 |
| | | | | |
| Low level (LL) | $< 2000$ | | | 2 or 3 |
| | $CBH_{max} - CBH_{min} > 1000$ m | | | 1 |
| | & $CBH_{max} < 2000$ m | | | |
| Middle level (ML) | $2000$–$7000$ | | | 2 or 3 |
| High level (HL) | $\geq 5000$ | | | 2 or 3 |
| | | | | |
| Multilayer uniform (MuUni) | | $< 0.5$ | $< 120$ | 2 or 3 |
| Multilayer transparent (MuTt) | | $> 0.5$ | $< 120$ | 2 or 3 |
| Multilayer patchy (MuPa) | | $> 0$ | $> 120$ | 2 or 3 |

cloud types of the second and third cloud layer could not be determined with the current algorithm. Hence, multilayered classes rather inform of the presence of other cloud layers on top of the lowest, classified, cloud layer.

Before analyzing the cloud type data produced by the algorithm, the performance of the algorithm was investigated. To test the performance, a third sample of 204 total sky images was selected, and the cloud type determined through visual inspection. These cloud types were compared with the results of the algorithm in matrix form (Table 2). The results showed that the overall performance of the algorithm was 68.4 %. The performance depended on the cloud type. Some clouds, such as nimbostratus, cause very distinguishable changes in solar radiation, and hence were easily determined by the algorithm, while

some other types, such as altocumulus and altostratus, cause similar changes as cirriform clouds, and were more often mixed by the algorithm. Indeed, the most often the algorithm mixed similar types of clouds, e.g. cumulus and stratus to stratocumulus.





**Table 2.** Contingency table presenting the performance of the cloud classification algorithm compared to the cloud types determined with visual inspection from total sky images.

| | | Algorithm | | | | | | | |
| | | Cumulus | Stratus | Stratocumulus | Nimbostratus | Altocumulus + Altostratus | Cirrus + Cirrocumulus + Cirrostratus | Clear + Cirrus | Other types | Agreement (%) |
|---|---|---|---|---|---|---|---|---|---|---|
| Visual inspection | Cumulus | **19** | 0 | 3 | 0 | 0 | 0 | 1 | 3 | 73 |
| | Stratus | 0 | **28** | 5 | 0 | 0 | 0 | 0 | 1 | 82 |
| | Stratocumulus | 0 | 7 | **10** | 0 | 0 | 0 | 0 | 2 | 53 |
| | Nimbostratus | 0 | 0 | 0 | **4** | 0 | 0 | 0 | 0 | 100 |
| | Altocumulus + Altostratus | 0 | 0 | 0 | 2 | **58** | 5 | 0 | 15 | 73 |
| | Cirrus + Cirrocumulus + Cirrostratus | 0 | 0 | 0 | 0 | 5 | **11** | 0 | 6 | 50 |
| | Clear + Cirrus | 0 | 0 | 0 | 0 | 0 | 0 | **15** | 6 | 71 |
| | Other types | 1 | 0 | 1 | 0 | 4 | 0 | 0 | **2** | 25 |

"Other types" includes cases when the cloud type was changing, and two types of clouds were present in the same image, or cases when it was hard for the observer to distinguish between two similar cloud types (e.g. stratocumulus and stratus).

The data presented in the (TR,PA) plane forms an upside down facing U-shaped pattern (Fig. 4). The physical reason for the
U-shape is as follows: when the transparency is high, the patchiness may be either low or high (clear sky vs. cumulus cloud streets), and when the transparency is low, the patchiness is also low (stratiform clouds). There are simply no clouds with low patchiness that would simultaneously have moderate transparency (0.6-0.8). Those transparency values are typical for cumulus clouds which are naturally patchy.

### 3.3 Cloud statistics using the new cloud algorithm

The algorithm was applied to the data from 2014 and 2016–2017 from SMEAR II. Only cases when SZA was less than 70° were included. Figure 5 displays the monthly occurrence of each cloud type with respect to the total number of data points in that month. Figure 5a-d represent the classified cloud types of the lowest cloud layer detected by the ceilometer. As clear sky cases and cases when the cloud class could not be determined are included (Fig. 5c and d), Fig. 5a-d will give the frequency of occurrence of each cloud type month-wisely. Therefore, summing the percentages corresponding to these classes (Fig. 300 5a-d) monthly will give 100 %. Figure 5e and f represent the second and the third cloud layer in multilayered cloud cases. Overall, the most commonly observed cloud types were stratus (28 %), cumulus (26 %) and stratocumulus (18 %), which altogether comprised approximately 70 % of clouds. Cirriform clouds were rarely observed, accounting only for about 2 % of the classified clouds. Clear sky and cirrus cases contributed 16 % of the classified cases.





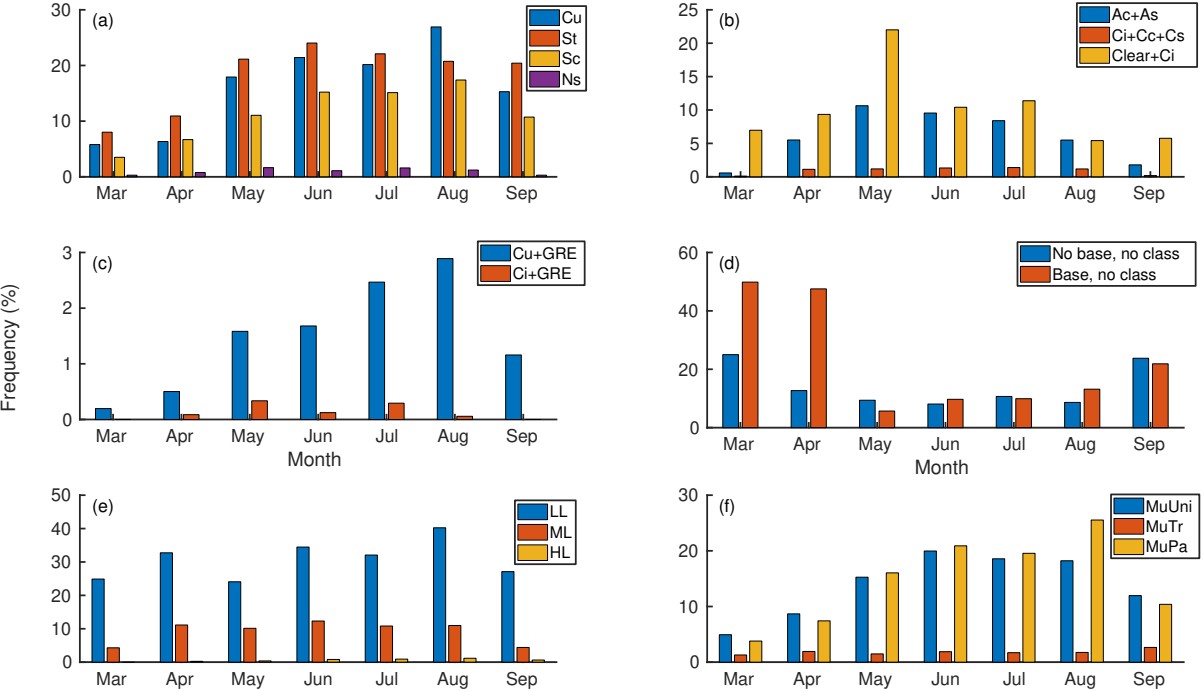

**Figure 5.** Occurrence of cloud types with respect to total number of data points month-wisely. (a) Low clouds, (b) middle and high clouds, (c) cumulus and cirrus clouds causing GRE, (d) cases when the cloud class could not be determined in situations when the ceilometer did or did not detect a cloud base, (e) height classes of multilayered clouds, and (f) characteristic classes of multilayered clouds. Notice the differences in y-axes scaling.

The seasonal variation of each cloud type was studied (Fig. 5). Many cloud types showed a robust seasonal variation. These cloud types had a maximum in occurrence during summertime, although the time of the maxima differed. Stratus, altocumulus and altostratus, and clear and cirrus classes had maxima in occurrence in early summer in May or June, while cumulus and stratocumulus had maxima in late summer in August. The seasonal variation of cumulus clouds causing global radiation enhancement followed the variation of cumulus cloud occurrence. Cirriform cloud occurrence did not show a clear seasonal variation, and the seasonal variation of cirrus clouds causing global radiation enhancement was also minor.

The relative share of "Base, no class" and "No base, no class" cases peaked in the beginning and end of the period of investigation (Fig. 5d). This indicates that the classification of clouds was more difficult in spring and autumn compared to summer. This may be caused by the fact that the used total sky images were taken between 1 May and 31 July, leading to over-representation of summertime clouds. Thus, the number of undefined cases could increase in spring and autumn. It should, however, be noted that wintertime was not included.

Figure 5e shows that the most commonly observed multilayered cloud type was the low level class (79 %). This is in line with the CBH observations (Fig. B4). Multilayered clouds with middle level cloud layer were also observed often (25 %),





whereas high level multilayered clouds were seldom observed (2 %). Overall, the relative fraction of high clouds was smaller compared to the other cloud types (Fig. 5a and b, 2 and B2).

Multilayered clouds were characterized by low transparency: 41 % of the clouds were patchy and 38 % uniform (Fig. 5f).
Multilayered clouds were transparent in 5 % of the cases. The reason why the numbers do not sum up to 100 % is the missing radiation data needed for the calculation of transparency and patchiness.

When multilayered clouds were present, the lowest cloud layer was most often stratus (29 %), stratocumulus (23 %) or cumulus (18 %). The first cloud layer was determined as low cloud in 72 % of the multilayered cloud cases. Hence, there was typically a low cloud layer above another low cloud layer. This can also be seen in Fig. B4 and B5 as the distance between
the cloud layers was usually less than 400 m. Accordingly, the seasonal variations of multilayered uniform and multilayered patchy clouds follow the seasonal variations of stratiform and cumulus clouds, having the maxima in early and late summer, respectively (Fig. 5f). The multilayered transparent cloud type did not show seasonal variation. Transparent clouds are cirriform clouds, and nor did they show seasonal variation. The lack of seasonal variation of cirriform clouds may partially be related to the relatively high occurrence of clear sky and cirrus cases in summertime, because also this class contains cirriform clouds as
they cannot be distinguished from the clear sky (Fig. 5b).

The diurnal variation of cloud types show that low cumulus clouds peak in the afternoons (Fig. B6a). Similar diurnal variation can also be seen in the frequency of low and patchy multilayered clouds (Fig. B6e and f). Stratus, nimbostratus, and altocumulus and altostratus were more common in the morning and evening compared to noon (Fig. B6a and b). As multilayered uniform and middle level layered clouds showed similar diurnal variation (Fig. B6e and f), those clouds were probably responsible for
the observed variation in multilayered clouds. Clear sky combined with cirrus clouds were most often observed in the morning (Fig. B6b). Global radiation enhancement during the presence of cumulus clouds were more common in late afternoons and evenings compared to mornings while during the presence of cirrus clouds it took place both in the mornings and evenings (Fig. B6c).

### 3.4   Brightness parameter

Brightness parameter ($P_\mathrm{B}$) is determined as a relation between the measured global radiation and the radiation at the top of the atmosphere averaged over half an hour:

$$P_\mathrm{B} = \frac{I_\mathrm{meas}}{I}. \tag{7}$$

The parameter has been used as a simplified measure of the prevailing cloudiness in terms of "cloudy" or "clear sky" (Kulmala et al., 2010, 2014a; Dada et al., 2017). In Kulmala et al. (2010), the limit of clear sky was set to $P_\mathrm{B} > 0.50$, in Kulmala et al.
(2014a) it was $P_\mathrm{B} > 0.60$, and in Dada et al. (2017) it was $P_\mathrm{B} > 0.70$. The limit of cloudy sky in all the three articles was $P_\mathrm{B} < 0.30$.

Figure 6 demonstrates the brightness parameter values obtained when different cloud types were present. We can see that even when the brightness parameter was above 0.7 (black line in Fig. 6), different types of clouds were present. Only stratus





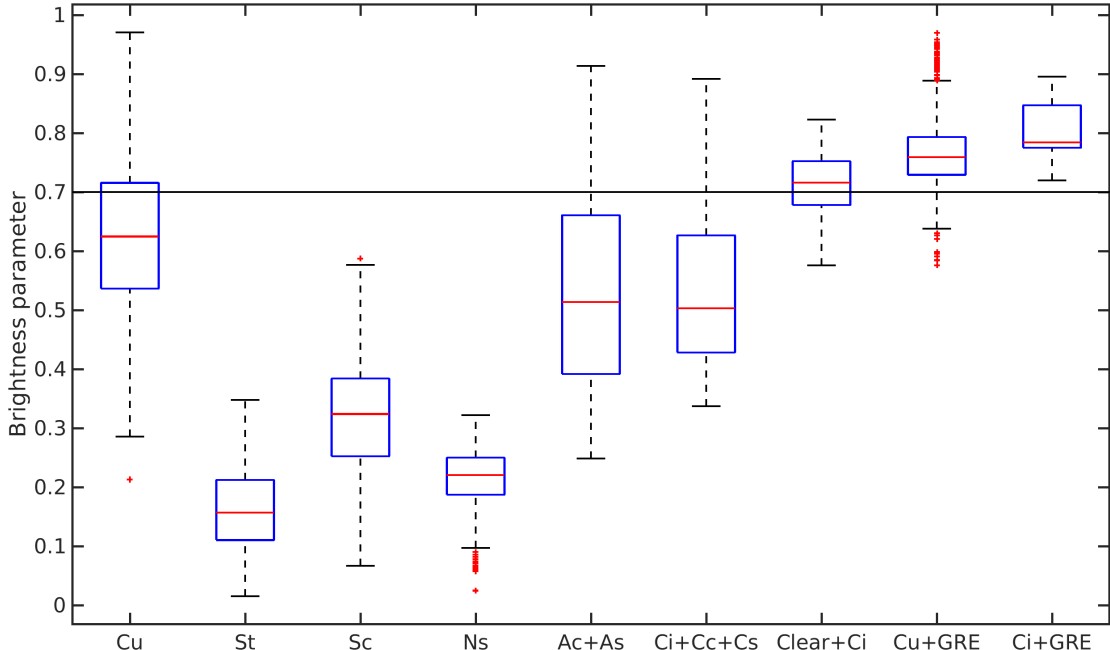

**Figure 6.** Obtained brightness parameter values when different types of clouds were present. The brightness parameter was calculated as 21 min running average. The values are from daytime (9:00–15:00) during the maximum growing season (from June to August). Red lines show the median values, lower and upper edges of the boxes are 25th and 75th percentiles, and whiskers correspond to 99.3 % coverage of the data. More extreme values are represented separately with red "+" symbol. The black line represents the limit for clear sky used e.g. in Dada et al. (2017).

and nimbostratus were not observed with the lowest brightness parameter limit ($P_B > 0.5$). According to our results, cumulus,
altocumulus, altostratus and cirriform clouds occurred when the brightness parameter was above the 0.6 or 0.7 limit.

The analysis of Dada et al. (2017) was related to aerosol formation. They concluded that aerosol formation was enhanced under clear sky conditions that were determined by the brightness parameter. Our findings indicate that clouds could be present during those days. A possible implication is that there could be a mechanism similar to that in the tropics where aerosol particles formed in the upper layers of the atmosphere are delivered to the surface by convective plumes that are often enhanced in the
presence of boundary layer clouds (Perry and Hobbs, 1994; Twohy et al., 2002; Waddicor et al., 2012; Leino et al., 2019; Lampilahti et al., 2020). Hence, cloudy cases falsely classified as clear sky might have complicated the analyses related to ecosystem–atmosphere interactions and new particle formation, hindering the understanding of the processes occurring in the boundary layer (Kulmala et al., 2010, 2014a; Dada et al., 2017). Our results show that a single parameter may not indicate clear sky conditions reliably, and thus when using brightness parameter in analysis, extra care should be taken when drawing





the conclusions. The new algorithm is an important tool in the future research regarding the radiation partitioning modified processes.

## 3.5  Discussion

Karlsson (2003), Pipatti et al. (2010), and Joro et al. (2010) reported similar frequency of cloud occurrence and yearly variations in Finland, based on satellite and surface observations, as we found in Hyytiälä from the ceilometer measurements (Fig. 2a).
The cloud occurrence retrieved with the pyranometer method were found to be higher than those measured with the ceilometer, and the seasonal variation was absent (Fig. 2b). One reason for the difference between ceilometer and pyranometer results might be the limited vertical resolution of the ceilometer, in which case some of the highest clouds would not be detected, lowering the observed cloud occurrence. As pyranometer measures only the attenuation of solar radiation, the altitude of the cloud does not affect its performance. However, this explanation is improbable because the cloud occurrence estimated using
the pyranometer method gave higher values than those found in the literature. Furthermore, the several studies mention the capability of ceilometers to detect clouds reliably, though their field of view is narrow, and performance is better with low clouds (Rodriguez, 1998; Kalb et al., 2004).

In Fig. 2, the cloud occurrence from the ceilometer observations contained also nighttime data points contrarily to the pyranometer data which were filtered by SZA. When only daytime (9:00-15:00) data from the time when SZA was less than
70° were used in both methods, the difference between the calculated cloud occurrences was reduced slightly (Fig. B7). The cloud occurrence estimated from the ceilometer measurements increased, presumably because the cloud occurrence had a maximum during daytime (Fig. B1). Additionally, the cloud occurrence from the pyranometer method decreased, implicating that the pyranometer method overestimated the cloudiness when data from early mornings and evenings were included, despite the filtering with SZA. Due to Finland's northern location, SZA is high throughout the year compared to locations closer to the
Equator. Hence, as the pyranometer method is sensitive to SZA, the most reliable results are obtained during the hours when the Sun is at the highest position, especially in summertime when cloud occurrence had a diurnal cycle (Fig. B1). Moreover, as shown also in Fig. 6, the simple limits set for determining cloudiness may not be efficient in all cases.

The cloud occurrence by the pyranometer was also modulated by the averaging over a 21 min time window. Thereby also cloudless data points might have been considered as cloudy whereas the ceilometer separated clear and cloudy periods.
However, the best practice to separate between clear and cloudy cases depends on the application. For example, if the objective is to quantify albedo, it is reasonable to rely on ceilometer data. Yet, if the objective is to study the effect of clouds generally on the ecosystem, pyranometer data averaged over 21 min are more appropriate in describing the integrated effect of changing light conditions on plants.

Despite the good agreement with the frequency of the cloud occurrence with values found in other studies, we are likely
to miss the occurrence of the second and third cloud layer (Fig. B2). Costa-Surós et al. (2013) found similar occurrence of multilayered clouds in Girona, Spain, using identical ceilometer. They compared their results with observations from the nearby airport, and noticed that the ceilometer overestimated the occurrence of single-layered clouds. They hypothesized that it might be due to the occultation of the laser pulse by the first cloud layer, and the fact that the vertical resolution of the





ceilometer was too low to detect all high clouds. The occultation by the first cloud layer might be an important phenomenon in

Hyytiälä where low and stratiform clouds were frequently observed (Fig. 5). Other studies have also shown higher frequencies of multilayered clouds (Wang and Rossow, 1995; Wang et al., 2000; Li et al., 2015), and differences in detection of cloud layers depending on the method (Wang et al., 1999, 2000; Rossow et al., 2005; Rossow and Zhang, 2010). Additionally, when multilayered clouds were observed, they predominately were two-layered, having two low cloud layers on top of each other (Fig. B4). Comparing the results with previous publications, this indicates that we miss middle and high clouds (Rossow et al.,

2005; IPCC, 2013; Li et al., 2015).

The high contribution of low single-layered clouds also modulates the observed CBHs. Joro et al. (2010) investigated cloudiness in Finland by combining satellite and ceilometer data. They found that in February low clouds dominated the CBH distribution while in August there was a second maxima around 2 500 m. We found a similar second maximum in April, May and June but in August the distribution decreased towards the end of higher CBHs (Fig. 3 and B4). The second maximum that we

found was around 1 600 m, i.e. at lower altitude compared to results by Joro et al. (2010). However, Joro et al. (2010) reported results from only two months whereas we had data from three years. Despite the high frequency of low clouds, our findings produce the distribution of CBHs similar to Wang and Rossow (1995) and Wang et al. (2000) who reported averaged CBH distribution of satellite and rawinsonde, respectively, data from many stations.

Our observation that there is often a low level cloud layer on top of low clouds explains the small difference between cloud

layers (Fig. B5). Wang and Rossow (1995) reported separation distances between cloud layers. They found that most often the separation distance was about 1 km whereas we found that the distance between two consecutive cloud layers was about 400 m. However, the results are not completely comparable due to different data analyzing procedures.

The cloud classification algorithm was able to produce the correct cloud type in about 70 % of the cases (Table 2). When other types of clouds than those that are classified by the algorithm were excluded, the performance was up to 84 %. The performance

was better with clouds having distinguishable effects on radiative conditions. For example, very opaque nimbostratus clouds, the algorithm identified correctly in 100 % of the cases. The least accuracy was obtained with cirriform clouds (50 %). This may be caused by the weaker detection of the high clouds by the ceilometer.

The performance of our algorithm was significantly better compared to the 45 % agreement of the original algorithm by Duchon and O'Malley (1999), and 45 % agreement of an other algorithm employing also solar radiation measurements (Calbó

et al., 2001). Moreover, when Calbó et al. (2001) reduced the number of cloud classes from nine to five, the classifier reached 58 % agreement with human-observed cloud classes. The performance of the new algorithm was approximately similar to the average performance of the reviewed cloud classification algorithms in Tapakis and Charalambides (2013). Our simple algorithm is based on measurements by two common instruments: pyranometer and ceilometer, and hence the good performance compared also to other, more sophisticated or expensive methods, is remarkable.

We found that low clouds were frequently observed (Fig. 5). When comparing the results with surface observations from Finland, we found that the algorithm produced approximately a similar frequency of occurrence and diurnal variation (Fig. 5 and B6) as the observations in Eastman and Warren (2014) and *Climatic Atlas of Clouds Over Land and Ocean* (available online at https://atmos.uw.edu/CloudMap/, last access: 10 January 2020; method explained in Hahn and Warren (2007)). However,





the frequency of middle, and especially high cirriform clouds, were up to tenfold smaller compared to the values in *Climatic*
*Atlas of Clouds Over Land and Ocean* (Fig. 5b). This can partly be explained by the fact that our clear sky class contained
also cirrus cloud cases. The better accuracy for low clouds was, however, likely caused by the limitations of the ceilometer to
observe high clouds as discussed above. Moreover, Li et al. (2015) reported that high and middle clouds often coexist with
other types of clouds. According to their results, at 60°N high clouds are often observed together with low or middle clouds.
We could not capture cases with many cloud layers because of the occultation of the laser pulse. When studying our results,
high level multilayered clouds were seldom observed but rather two low cloud layers coexisted (Fig. 5).

Many cloud types showed seasonal variation, having a maximum in summertime, e.g. cumulus clouds peaking in late sum-
mer (Fig. 5). The algorithm reproduced the seasonal variation of clouds reported in *Climatic Atlas of Clouds Over Land and
Ocean*. However, as our analysis does not cover winter months, some possible discrepancies were observed: according to
*Climatic Atlas of Clouds Over Land and Ocean*, the occurrence of nimbostratus has a minimum in summertime, and the oc-
currence of stratus has a maximum in autumn. In our study, nimbostratus showed relatively constant frequency of occurrence
from April to August but was almost absent in March and September. Stratus had a maximum in June but the differences in
frequency with August and September were minor (Fig. 5a). Hence, we cannot conclude the deviation from the cloud obser-
vations reported in *Climatic Atlas of Clouds Over Land and Ocean*, and overall the performance of the simple algorithm was
very good.

**4   Conclusions**

The present study included a formulation of a cloud type classification algorithm, and investigation of cloud properties at
SMEAR II measurement site in Hyytiälä, Finland. The overall cloud occurrence measured by the ceilometer was in agreement
with the reported values in literature, though the frequency of single-layered clouds were likely overestimated, and the occur-
rence of middle and high clouds underestimated. We hypothesize that this is caused by the facts that the vertical maximum
measurement height of the ceilometer did not allow it to detect all the high clouds, and that the occultation of the laser pulse
by the lowest cloud layer prevented the observation of other cloud layers.

The developed cloud classification algorithm is based on two variables measured continuously at the station: global radiation
and CBH. Despite the simplicity of the algorithm, it can identify seven different cloud types along with classification of
multilayered clouds based on their base height and characteristics (uniform, transparent or patchy). The overall performance
of the algorithm was almost 70 %, indicating a good ability to distinguish cloud types observed over a boreal forest. The
algorithm may, however, be utilized also in other environments. Because the algorithm is based on attenuation of solar radiation,
the performance is better with cloud types that have a distinguishable impact on radiative conditions on the Earth, such as
nimbostratus. We are confident that the algorithm is able to reproduce the cloud types rather reliable in common situations,
though it is probable that it does not reproduce all the high and multilayered clouds due to the limitations of the performance
of the ceilometer, as discussed above. Indeed, we showed that low and optically thick stratiform and cumulus clouds occurred
frequently, indicating the high probability for occultation of the laser pulse.



The brightness parameter is defined as a ratio between the measured global radiation and calculated radiation at the top of the atmosphere, and has been used as an indicator of clear sky or presence of clouds. We found that cumulus, altocumulus, altostratus and cirriform clouds were present when brightness parameter indicated clear sky conditions. Thus, the studies

defining clear sky cases based on brightness parameter, may be biased. The new algorithm may be utilized in the future to distinguish clear sky conditions in a more reliable way.

As the focus of this study was in the development of the algorithm, we used data only from three years. The measurements of the CBH and the global radiation at SMEAR II have been ongoing since 2008, and therefore the analysis can easily be extended in the future for longer time periods and different data sets. The current algorithm is the first one indicating the

prevailing cloud types at SMEAR II, and we encourage to use it in studies related to the boundary layer interactions involving radiation processes and clouds.

*Data availability.* The used data measured at SMEAR II can be accessed with Smart-SMEAR online tool (https://avaa.tdata.fi/web/smart, last access: 09 January 2020, (Junninen et al., 2009)). AOD and precipitable water were obtained from AERONET data base are available at https://aeronet.gsfc.nasa.gov/ (last access: 04 February 2020, (Holben et al., 1998)). Total sky images are available at https://adc.arm.gov/

discovery/#v/results/s/fsite::tmp (last access: 11 March 2020, (Petäjä et al., 2016)). The cloud classification produced in this study is available upon request from the first author at ilona.ylivinkka@helsinki.fi.





# Appendix A: Materials and methods

## A1 Site and data set

When comparing the performance of the cloud algorithm using SMEAR II and ARM ceilometers, the results are similar on a
daily scale (Fig. A1). However, the distance between the ceilometers led to different results if data from certain moments of time
were examined. Thus, the classification algorithm developed with one instrument is also applicable with another instrument.

**Figure A1.** Comparison of the frequencies of the produced cloud types with the ceilometer of SMEAR II (blue) and ARM campaign (red)
during four random days. The frequencies were obtained by dividing the number of cloud type records by the total number of data points in
the day. The abbreviations of the cloud types are found in Table 1. Note the different limits of the y-axes.





# Appendix B: Results and discussion

## B1 Cloud properties

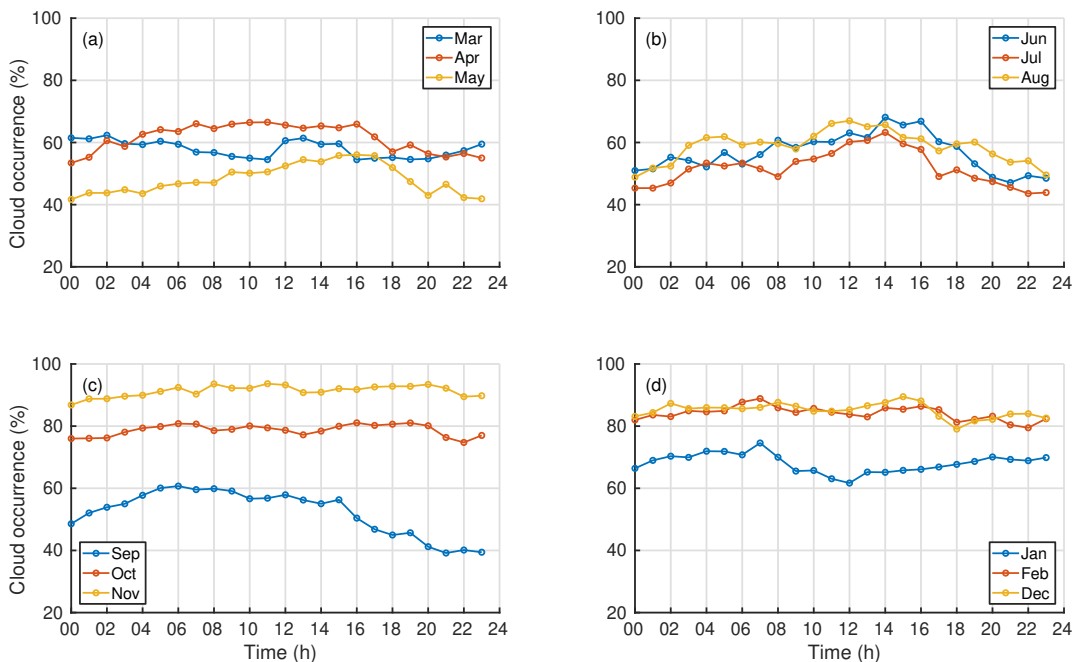

**Figure B1.** Diurnal variation of the cloud occurrence in (a) spring, (b) summer, (c) autumn and (d) winter months. The figure contains data from the lowest cloud layer measured with the ceilometer. The number of cloud observations was divided by the total number of data points in the certain hour to obtain the relative cloud occurrence.





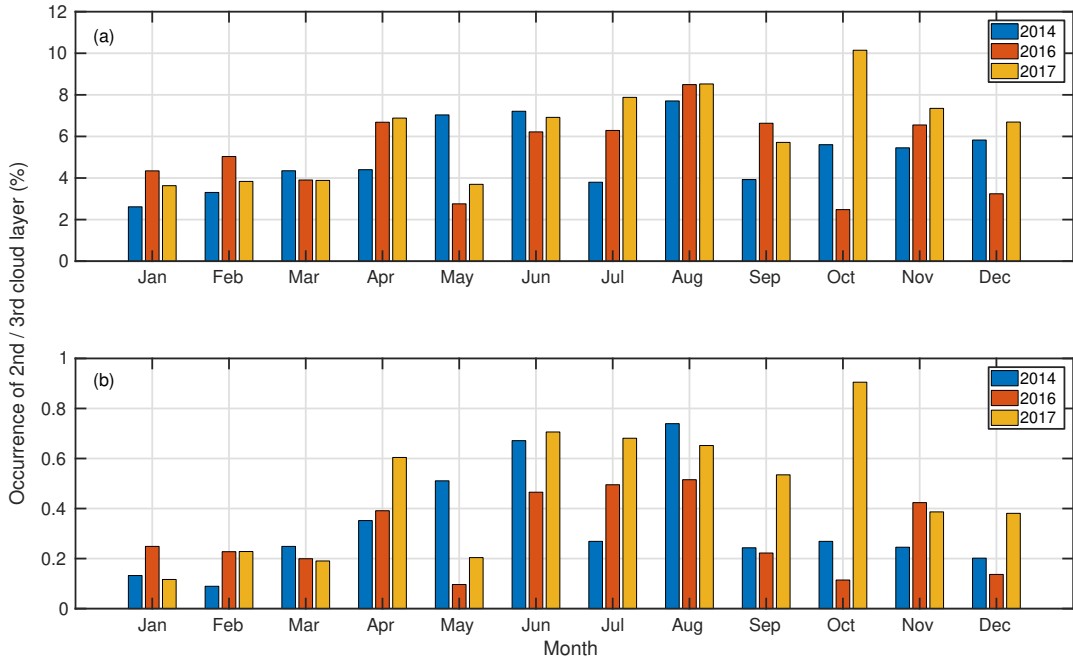

**Figure B2.** Monthly average occurrence of (a) the second and (b) the third cloud layer over Hyytiälä. Note the different limits of the y-axes.



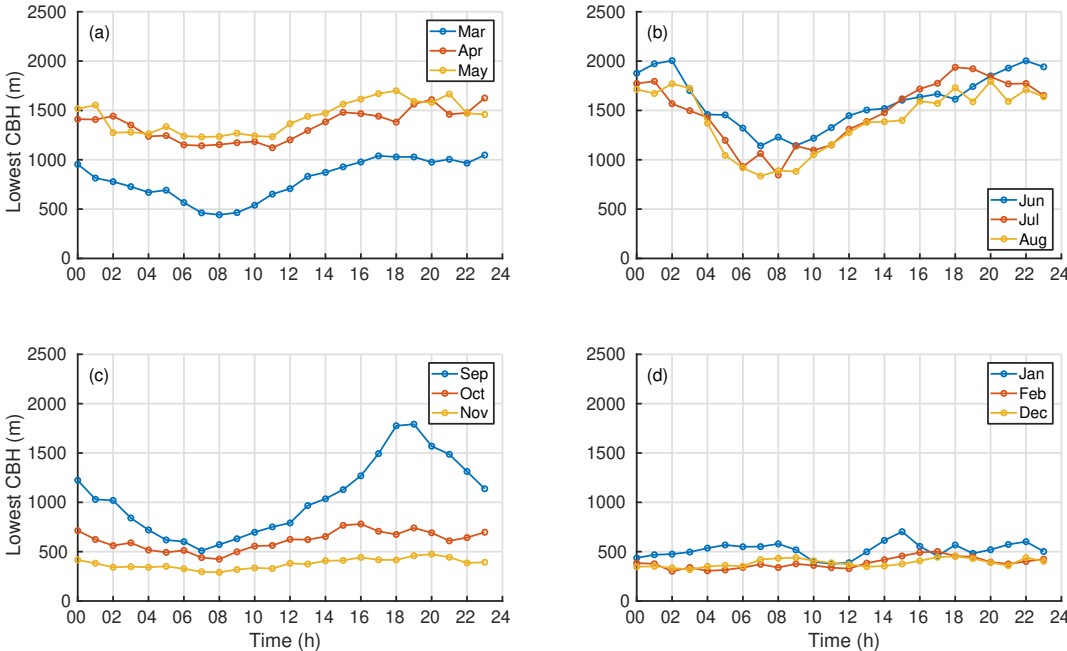

**Figure B3.** Hourly averaged CBH of the lowest cloud layer from (a) spring, (b) summer, (c) autumn and (d) winter months. Before calculating the hourly averages, the data from all three years was first separated month wisely.





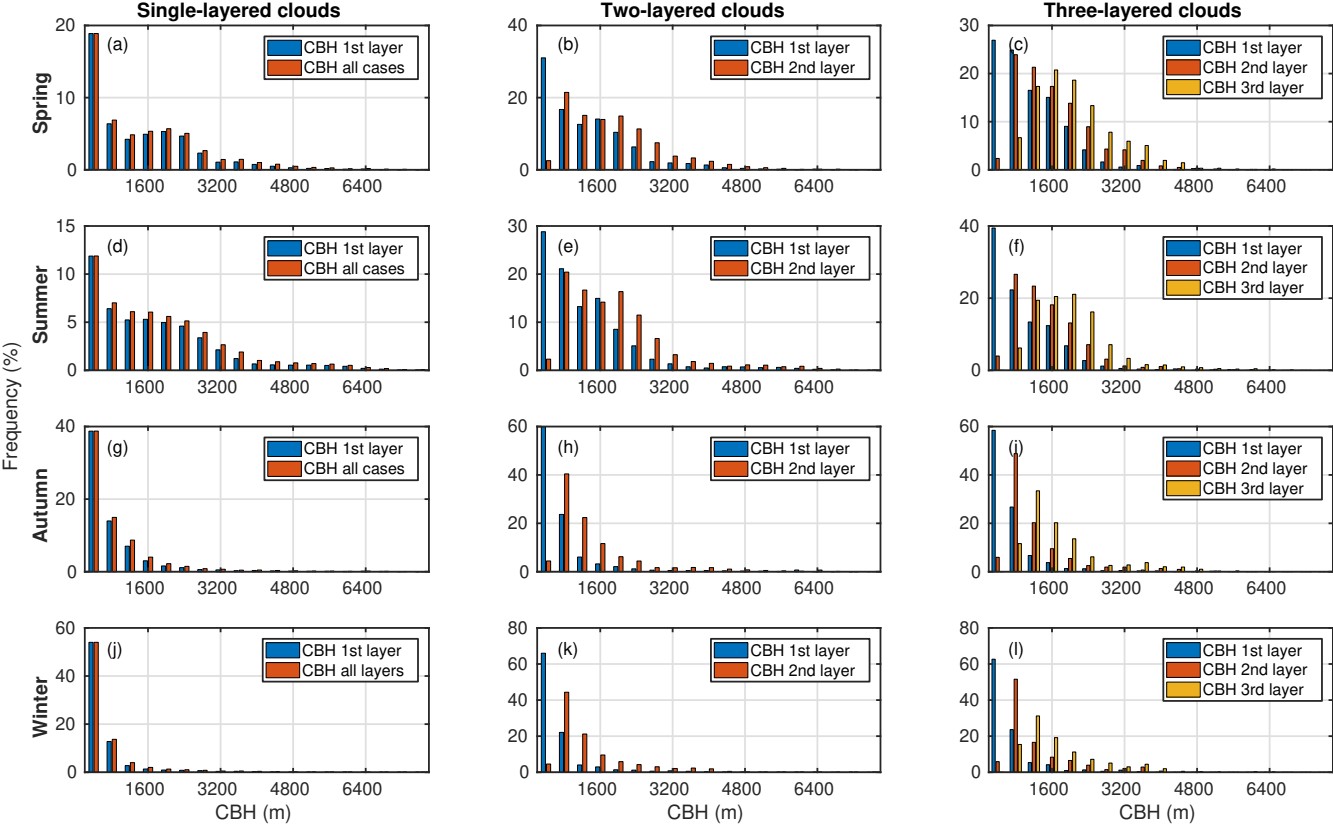

**Figure B4.** The frequency of single-layered clouds (a,d,g,j), two-layered clouds (b,e,h,k), and three-layered clouds (c,f,i,l). The CBH records of one season were divided into 400 m bins, and the frequencies were obtained by dividing the number of CBH records in each bin by the total number of CBH records in the season. Figures (a-c) represent spring, (d-f) summer, (g-i) autumn and (j-l) winter. Note the different limits of the y-axes.





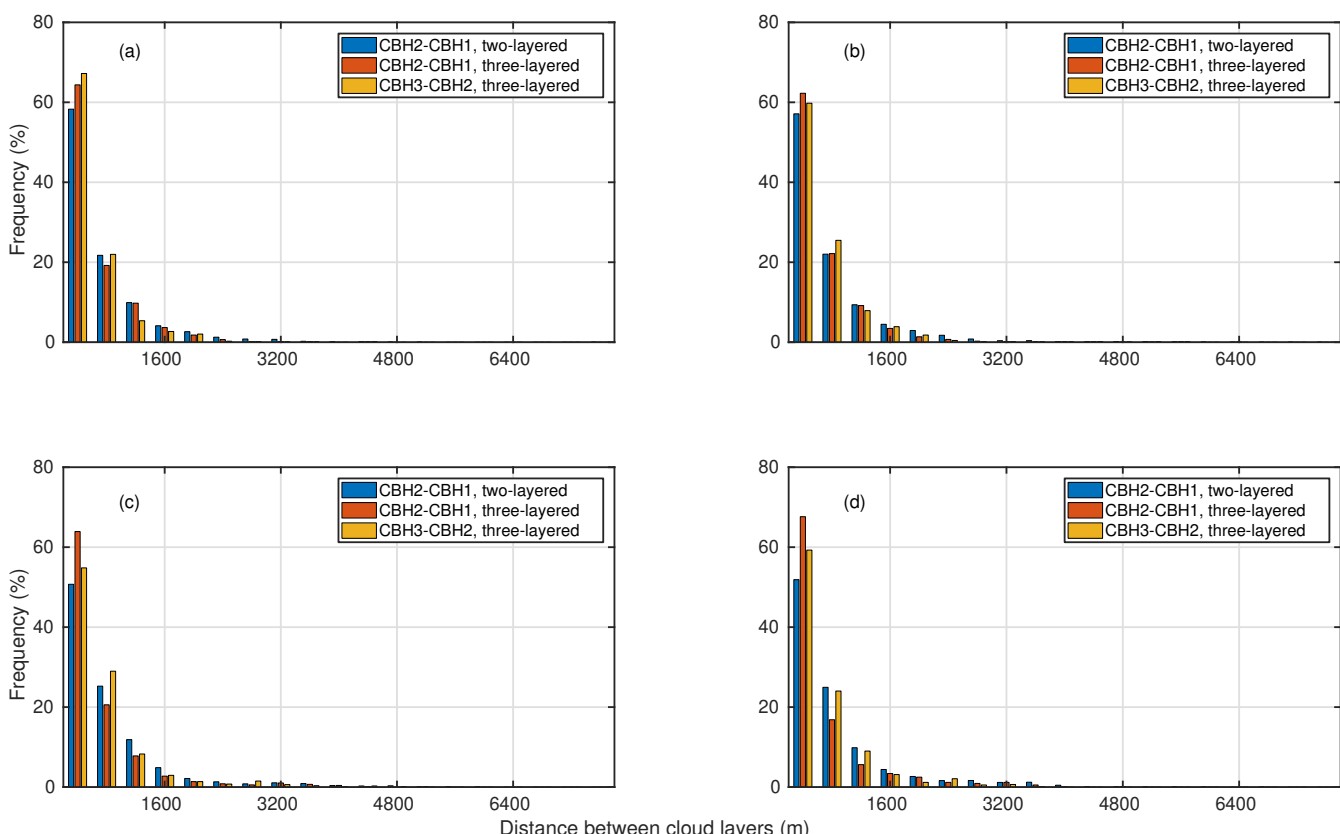

**Figure B5.** Distribution of the height difference between cloud layers. Blue represents the difference between the lowest and the second cloud layer in two-layered clouds, red similarly but in three-layered clouds, and yellow represents the difference between the second and the third cloud layer. (a) Spring, (b) summer, (c) autumn and (d) winter.





## B2 Cloud statistics using the new cloud algorithm

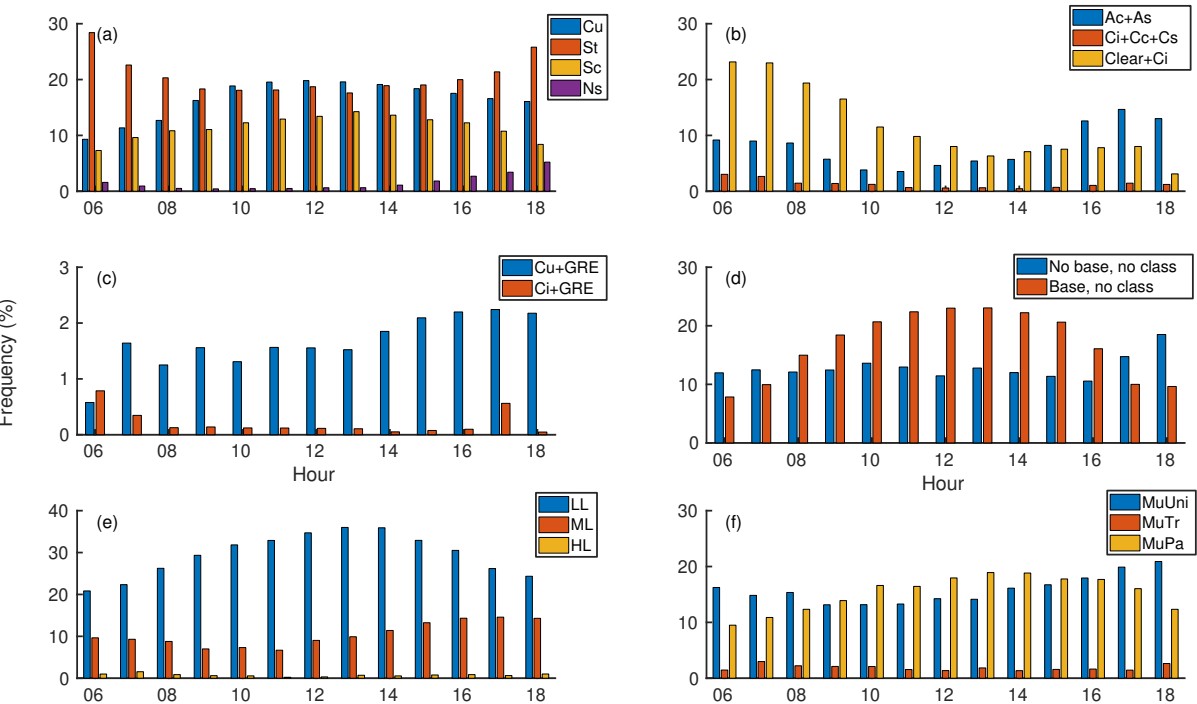

**Figure B6.** Diurnal variation of clouds types. (a) Low clouds, (b) middle and high clouds, (c) height classes of multilayered clouds, (d) characteristic classes of multilayered clouds, (e) cumulus and cirrus clouds causing GRE, and (f) cases when cloud class could not be determined in situations when the ceilometer did or did not detect a cloud base. The explanations of the abbreviations of the cloud classes are provided in Table 1. Notice the differences in y-axes scaling.





**B3    Discussion**

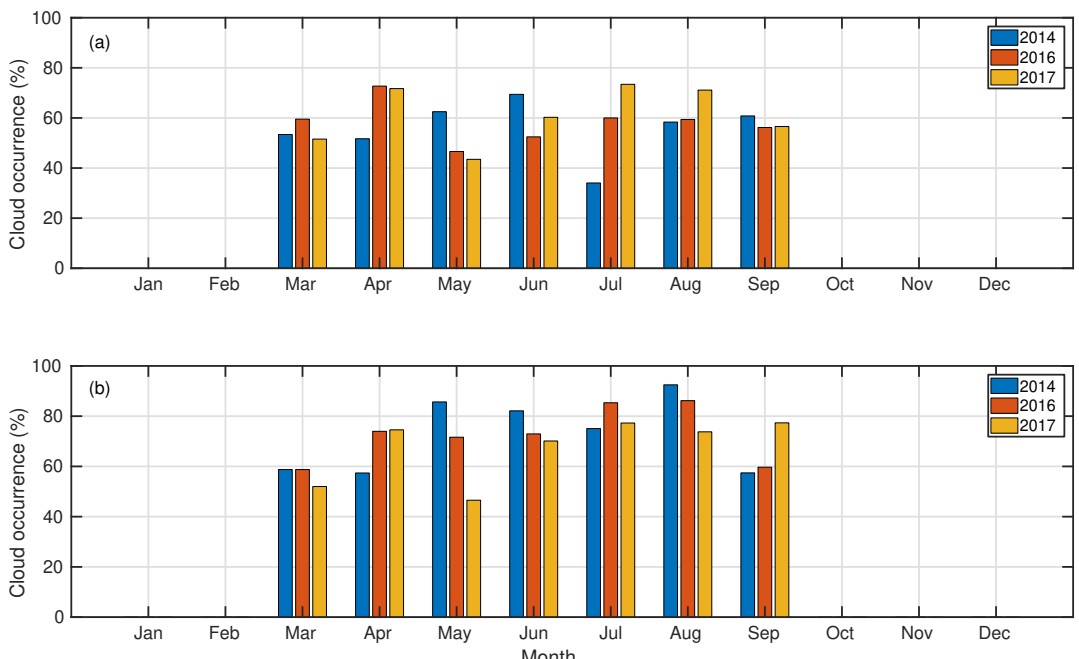

**Figure B7.** Monthly average cloud occurrence observed in Hyytiälä based on (a) the ceilometer and (b) the pyranometer measurements. Daytime (9:00–15:00) values from March to September when SZA was less than 70° were used. The number of cloud observations was divided by the total number of data points in one month to obtain the cloud occurrence. The occurrence estimated from the ceilometer measurements (64 %) was lower compared to the occurrence estimated from the pyranometer data (71 %).



*Author contributions.* EE and MK developed the concept of the study. MP and MV initiated the data analysis and development of the algorithm. SK refined the parameter ranges of the algorithm using total sky images, and estimated the performance of the algorithm. IY performed the data analysis for the paper and wrote it. EE, DT and VMK helped with the interpretation of the results. All authors gave critical feedback and helped with shaping the paper.

*Competing interests.* The authors declare that they have no conflict of interest.

*Acknowledgements.* This work was supported by the Academy of Finland Center of Excellence programme (grant no. 307331) and the Academy of Finland professor grant to Markku Kulmala (grant no. 302958). The results are part of a project (ATM-GTP/ERC) that has received funding from the European Research Council (ERC) under the European Union's Horizon 2020 research and innovation programme (grant agreement no. 742206). D. Taipale was supported by the Academy of Finland (no. 307957).



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
