# Peer review of "Clouds over Hyytiälä, Finland: an algorithm to classify clouds based on solar radiation and cloud base height measurements"

_Atmospheric Measurement Techniques, 2020_

## Referee Comment (RC1) · Anonymous Referee #1 · 15 Jun 2020

The classification and quantification of clouds from routine surface and remote-sense measurements remains essential information from studies that range from weather, atmospheric chemistry and the interaction between land and atmosphere. In this research, the authors present an algorithm that enables a cloud classification based on global radiation (observed with a pyranometer) and cloud base height (measured with a ceilometer). The algorithm is based on the calculation of three variables related to the cloud characteristics: transparency, patchiness and the measured CBH. By combining these metrics they are able to identify and classify low, middle and high clouds. To evaluate the performance of the algorithm thy compare with an observed who use total sky images. The agreement is 70 %. The paper explained and discussed very interesting

findings that can help the SMEAR II site -a referent site in the boreal ecosystem due to its completeness in measurements- and other sites. The article is very well written with a very complete introduction that stresses the relevance to have this sort of classification and quantification as a routine product for meteorological/atmospheric chemistry sites. The paper discussed interesting histograms of the cloud types monthly and daily averages, and as the authors mentioned in the conclusions, it will become a valuable tool to study the interactions between surface and the cloudy-boundary layer at boreal ecosystems. I agree with them. My comments to the article are the following:

1) For the completeness of the article, I would have appreciated a short section discussing the evaluation of the algorithm against satellite measurements. For instance the Meteosat Second Generation provides information on cloud classification. Please note that I am not asking a full comparison of the three years under analysis, but perhaps some case examples, for instance related to the diurnal variability or the more demanding and difficult to classify multi-layer clouds. Would it be possible to include this information?

2) Section 2.2 The length of the time interval (21 minutes) to calculate the transparency needs to be better justified. There is not a clear explanation on why it is used (only a reference to the work of Duchon and O'Malley (1999)). Is it related to a life time of clouds? More important, What is the sensitive of the algorithm to this value to the proposed classification? At the discussion, there is a short discussion on these values (lines 383-389), but it does not include the sensitivity to it.

3) Due to the completeness of the SMEAR II data set, I think it will be nice to attempt to connect the proposed metrics to other variables that are very relevant in the modelling of the clouds, but remain difficult to be measured. For example, Have the authors compared the transparency with an estimation of the cloud optical depth?

4) Equations (1) and (2). How do they model the clear sky radiation?

---

## Referee Comment (RC2) · Anonymous Referee #2 · 5 Jul 2020

This paper revisits the relatively old topic of guessing cloud characteristics from solar radiation measurements. Specifically in this case, the main novelty is the simultaneous use of ceilometer (cloud base height) data so in principle a better estimation of cloud type can be made. The paper is in general correct, but with some effort it could be quite significantly improved.

1. In my opinion, mixing the presentation and validation of the algorithm with "climatic" style (but for only 3 years) analysis of observations is somewhat confusing. So section 3.1 and then 3.3 and so, are kind of distracting the attention. I would focus on the new algorithm, so after sections 1 and 2 I would jump to current section 3.2. Then, you

could add a whole new section regarding results of applying the "occurrence" criteria and the new algorithm.

2. To my understanding, the fundaments used to determine occurrence (lines 177-78, "the ratio between the measured global radiation and modeled radiation at the top of the atmosphere (I)") is almost the same as the "brightness parameter" (lines 340-341, "relation between the measured global radiation and the radiation at the top of the atmosphere"). So, I would suggest defining this once, and then using for occurrence (setting a threshold) and after that using it also to further explore cloud type characteristics. Moreover, brightness parameter is usually known as "clearness index" in this context of cloud determination from solar radiation measurements. This would simplify the whole manuscript.

3. The dataset used to develop (and test) the algorithm is quite limited. Actually, the authors already recognize this (lines 312-13 "This may be caused by the fact that the used total sky images were taken between 1 May and 31 July, leading to overrepresentation of summertime clouds. Thus, the number of undefined cases could increase in spring and autumn"). In addition, the authors decided not to use observations with SZA > 70 deg. I understand their concerns, but this threshold is usually set at 80 deg. In summary, admitting observations up to 80 deg SZA, and using more whole sky images, would allow a largest number of cases to be used in the algorithm development and validation.

Besides these three general comments, that should be addressed as comprehensively as possible, I do have a number of minor comments:

a. L. 14. "aerosol formation" or simply "aerosol load"

b. L. 53-57. This paragraph breaks the introduction. In my opinion, it could appear later, along with the content of the paragraph starting in L. 84.

c. L. 84-89. In these 6 lines, the reference Duchon and O'Malley appears 4 times (!). I
understand that this reference is important in the present study, but this is made clear by saying that the new algorithm is based on that previous work. You don't need to repeat it 4 times.

d. L. 107, in the context of an applied Meteorology paper, the age of the pines is not relevant. Actually, it would be more relevant the height of trees and possible "shadows" on the instruments.

e. L. 118-119. You mention here that the Solis model is fed with AOD and precipitable water from Aeronet. This is ok, but then, this means that the method is not as easily "transferable" to other sites: they need to have not only pyranometer and ceilometer, but an Aeronet (Cimel sunphotometer) equipment too.

f. L. 124-125. Authors introduce here, too early, the idea of "parameter ranges". I think is not needed yet, before presenting the method for cloud classification.

g. L. 141-142. I would say that the ceilometer is not at all (or hardly, if any) sensitive to SZA.

h. L. 156. Why not presenting and discussing here the need for "scaling" radiation measurements?

i. L. 157. I would start a new paragraph regarding TR_max. Otherwise, it might be confused with the CBH from ceilometer.

j. L. 159. Well, before applying the classification you don't know if there are cumulus clouds. You should mention any type of varying cloudiness (obscuring and not obscuring the sun).

k. L. 178. You could better use "irradiance on a horizontal surface" instead of the too generic "radiation".

l. Section 2. Is cloud occurrence from ceilometer data described in this methods section?

m. L. 200-201. Cloud occurrence from radiation data cannot show seasonality as 5 moths are missing.

n. L. 224. How are middle and high clouds distinguished if there is a range (5000-7000 m) where both are cloud types are included?

o. L. 241. What do you mean by "uniformly and randomly"?

p. L. 244. "We first VISUALLY classified..."

q. Figure 4, caption. Explain the meaning of whiskers.

r. L. 289. With the current form of Fig. 4, the U shape is difficult to catch.

s. L. 341. "averaged over half an hour", but later you talk about 21 minutes. Please clarify (and consider my previous general comment #2)

t. Section 3.4, last paragraph. This is a somewhat confusing paragraph. If this paper is about cloud classification, why a discussion about ecosystem interactions? And, in this section the algorithm is not used, but the brightness parameter. Regarding the use of 0,7 as threshold, it is clearly too low to garantee a clear sky. In any case, if this is a discussion about results, it would fit better in section 3.5.

u. L. 464-465. Well, this is a matter of which threshold you use. The "clearness index" in different versions is used to detect clear skies, by applying (if I'm right) a higher threshold (about 0,9).

---

## Author Response (AR1)

Review of Ylivinkka et al. "Clouds over Hyytiälä, Finland: an algorithm to classify clouds based on solar radiation and cloud base height measurements" by Anonymous Referee #1

We thank Referee 1 for the valuable comments and improvements to our manuscript. We have revised our manuscript, and provide below a point-by-point answers to the comments, which are repeated in italic.

*The classification and quantification of clouds from routine surface and remote-sense measurements remains essential information from studies that range from weather, atmospheric chemistry and the interaction between land and atmosphere. In this research, the authors present an algorithm that enables a cloud classification based on global radiation (observed with a pyranometer) and cloud base height (measured with a ceilometer). The algorithm is based on the calculation of three variables related to the cloud characteristics: transparency, patchiness and the measured CBH. By combining these metrics they are able to identify and classify low, middle and high clouds. To evaluate the performance of the algorithm thy compare with an observed who use total sky images. The agreement is 70 %. The paper explained and discussed very interesting findings that can help the SMEAR II site -a referent site in the boreal ecosystem due to its completeness in measurements and other sites. The article is very well written with a very complete introduction that stresses the relevance to have this sort of classification and quantification as a routine product for meteorological/atmospheric chemistry sites. The paper discussed interesting histograms of the cloud types monthly and daily averages, and as the authors mentioned in the conclusions, it will become a valuable tool to study the interactions between surface and the cloudy-boundary layer at boreal ecosystems. I agree with them. My comments to the article are the following:*

We are grateful for the positive viewing of our manuscript and helpful comments to improve it.

*1) For the completeness of the article, I would have appreciated a short section discussing the evaluation of the algorithm against satellite measurements. For instance the Meteosat Second Generation provides information on cloud classification. Please note that I am not asking a full comparison of the three years under analysis, but perhaps some case examples, for instance related to the diurnal variability or the more demanding and difficult to classify multi-layer clouds. Would it be possible to include this information?*

We thank Referee for a valuable comment. Data from satellite products could surely be used as an optional parameter to improve the algorithm, and especially its ability to detect and classify multilayered and high clouds. Additionally, we could test how well satellite cloud classification and satellite-derived parameters with clearness index and patchiness compare with the results of our algorithm. This is, however, something that we must investigate further in the future. Now, we tested five random case examples of satellite images taken over southern Finland against the classification made by the algorithm. The results are shown below. The selected days were (a) 03 May 2016, (b) 13 May 2016, (c) 27 May 2016, (d) 07 June 2017 and (e) 17 June 2017. BNC refers to the "Base, no class" and NBNC to the "No base, no class". We used satellite data provided by NASA Worldview (https://worldview.earthdata.nasa.gov/, last access: 27 August 2020). We can see that mostly the algorithm was able to produce similar clouds as seen in the satellite image.

[Figure]

[Figure]

*2) Section 2.2 The length of the time interval (21 minutes) to calculate the transparency needs to be better justified. There is not a clear explanation on why it is used (only a reference to the work of Duchon and O'Malley (1999)). Is it related to a life time of clouds? More important, What is the sensitive of the algorithm to this value to the proposed classification? At the discussion, there is a short discussion on these values(lines 383-389), but it does not include the sensitivity to it.*

We added a new paragraph to better describe the use of 21 min interval. The text now reads (L. 157):
"The chosen time interval in this work was 21 min, similar to Duchon and O'Malley (1999) to be able the compare our results. However, the length of the time interval is based on empirical experience of the time span of cloud variability in the sky and the life time of clouds. Cumulus clouds are the largest patchy clouds, and hence they are used as a reference for the time span of clouds. The representative size of typical cumulus clouds is 1 km and if assuming that the average wind speed is about 3–6 m s $^{-1}$ (Stull, 2000), then during 21 min the clouds can move 3.8–7.6 km, meaning that roughly 4–8 clouds can pass the measurement beam of the instruments. Capturing several clouds is necessary for the calculation of standard deviation, which is employed when calculating patchiness as described below. Hence, decreasing the 21 min time interval can be problematic due to insufficient number of passing clouds, needed for the calculation. Moreover, a study by Rodts et al. (2003) show that ca. 1 km sized clouds dominate the vertical mass and buoyancy fluxes. Thus, they can be expected to be optically thicker than smaller or larger

clouds, and thereby they cause the largest decrease in solar radiation which contributes to the standard deviation the most. Rodts et al. (2003) also showed that the cloud cover density is dominated by intermediate clouds with linear sizes of 0.7–1 km. This means that they give the largest contribution to the cloud cover, determined as a ratio of the 2D projection of the area occupied by clouds to the total image area. Another time constraint is related to the life time of clouds. A typical life time of cumulus cloud is 20 min, so 21 min is a reasonable to capture one life cycle of cumulus clouds (Lohmann et al., 2016). Other clouds have longer life times (Lohmann et al., 2016). Therefore, we can expect that 20–30 min would give the same results but considerably shorter time intervals would not give the best representation of the overall cloudiness conditions and longer time interval will increase the number of poorly defined cases when there is a transition from one type of cloudiness to another."

*3) Due to the completeness of the SMEAR II data set, I think it will be nice to attempt to connect the proposed metrics to other variables that are very relevant in the modelling of the clouds, but remain difficult to be measured. For example, Have the authors compared the transparency with an estimation of the cloud optical depth?*

We thank Referee for pointing out this important question related to connection of modeling and measurement results. In Hyytiälä cloud optical depth (COD) has been measured with Three-Waveband Spectrally-agile Technique (TWST) sensor (Niple and Scott, 2016) during the BAECC campaign. Transmittance is related with COD with a theoretical formula (Sena et al., 2016, Eq. (6)). We plotted the transmittance against measured COD along with the theoretical relation curve, and hence compared the results. We could conclude that COD for non-patchy clouds can be estimated utilizing transmittance during daytime. We added new sections 2.4 and 3.4.2 to describe the theory and results related to COD.

*4) Equations (1) and (2). How do they model the clear sky radiation?*

We clarified the behavior of transmittance and patchiness in clear sky and cloudy conditions (L. 151):
"Transmittance is the ratio of the measured global radiation (I_meas) to the modeled clear sky radiation (I_gh), given by Eq. (5), averaged over a running time interval: ...", (L. 155):
"Transmittance describes how effectively clouds block solar radiation. It is equal to 1 in clear sky conditions and approaches 0 for an overcast sky.", and (L. 177):
"The modeled clear sky radiation is calculated using Eq. (5). Patchiness determines the variability of the cloud layer. The value is smallest both for uniform and overcasting, and clear sky conditions, and increases in partly cloudy conditions."

Besides the comments by Referee, we changed term "transparency" to more generally used "transmittance". We additionally removed parameter "TR_max" as in the further examination of the algorithm it was found to be redundant. Because the data availability of AERONET data was low during March, April and September of 2016 and 2017, we calculated median values of the available data separately for spring (March and April) and September, and applied those when data was missing in those months. Lastly, we changed the upper limit of transmittance for stratus clouds from 0.4 to 0.6. This was done because we could see that many St clouds fall into this area but were previously not classified. These latter two changes to the algorithm decreased the number of cases

falling into the class "Base, no class". Now also the frequency of occurrence of stratus clouds is better in accordance with observations in *Climatic Atlas of Clouds Over Land and Ocean* (available online at https://atmos.uw.edu/CloudMap/, last access: 10 January 2020). We additionally changed the upper transmittance limit of Ns from 0.4 to 0.3 and lower transmittance limit of Ac+As clouds from 0.4 to 0.3. This was done because we could see that especially in springtime Ac+As clouds were previously falsely classified as Ns. The overall occurrence of Ns clouds decreased (from 1.4 % to 0.6 %) and occurrence of Ac+As increased (from 9.0 % to 10.3 %) but otherwise the change did not affect our results.

**References**

Duchon, C. E. and O'Malley, M. S.: Estimating cloud type from pyranometer observations, Journal of Applied Meteorology, 38, 132–141, 1999.

Lohmann, U., Lüönd, F., and Mahrt, F.: An introduction to clouds: From the microscale to climate, Cambridge University Press, 2016.

Niple, E. R. and Scott, H. E.: Biogenic Aerosols – Effects on Climate and Clouds. Cloud Optical Depth (COD) Sensor Three-Waveband Spectrally-Agile Technique (TWST) Field Campaign Report, Tech. rep., United States, https://doi.org/10.2172/1248494, 2016.

Rodts, S. M. A., Duynkerke, P. G., and Jonker, H. J. J.: Size Distributions and Dynamical Properties of Shallow Cumulus Clouds from Aircraft Observations and Satellite Data, Journal of the Atmospheric Sciences, 60, 1895–1912, https://doi.org/10.1175/1520-0469(2003)060<1895:SDADPO>2.0.CO;2, 2003.

Sena, E. T., McComiskey, A., and Feingold, G.: A long-term study of aerosol–cloud interactions and their radiative effect at the Southern Great Plains using ground-based measurements, Atmospheric Chemistry and Physics, 16, 11 301–11 318, https://doi.org/10.5194/acp-16-11301-2016, 2016.

Stull, R. B.: Meteorology for scientists and engineers, Brooks/Cole, 2nd edn., 2000.

Review of Ylivinkka et al. "Clouds over Hyytiälä, Finland: an algorithm to classify clouds based on solar radiation and cloud base height measurements" by Anonymous Referee #2

We thank Referee 2 for the valuable comments and improvements to our manuscript. We have revised our manuscript, and provide below a point-by-point answers to the comments, which are repeated in italic.

*This paper revisits the relatively old topic of guessing cloud characteristics from solar radiation measurements. Specifically in this case, the main novelty is the simultaneoususe of ceilometer (cloud base height) data so in principle a better estimation of cloud type can be made. The paper is in general correct, but with some effort it could be quite significantly improved.*

We are grateful for the positive viewing of our manuscript and helpful comments to improve it.

*1. In my opinion, mixing the presentation and validation of the algorithm with "climatic"style (but for only 3 years) analysis of observations is somewhat confusing. So section 3.1 and then 3.3 and so, are kind of distracting the attention. I would focus on the new algorithm, so after sections 1 and 2 I would jump to current section 3.2. Then, you could add a whole new section regarding results of applying the "occurrence" criteria and the new algorithm.*

We agree, and thank Referee for the valuable comment. We changed the order of the sections, and now Sect. 2 is followed by the section describing the derivation of the algorithm.

*2. To my understanding, the fundaments used to determine occurrence (lines 177-78, "the ratio between the measured global radiation and modeled radiation at the top of the atmosphere (I)") is almost the same as the "brightness parameter" (lines 340-341, "relation between the measured global radiation and the radiation at the top of the atmosphere"). So, I would suggest defining this once, and then using for occurrence(setting a threshold) and after that using it also to further explore cloud type characteristics. Moreover, brightness parameter is usually known as "clearness index" in this context of cloud determination from solar radiation measurements. This would simplify the whole manuscript.*

Referee is correct. We unified the terminology and theory related to cloud occurrence calculations from pyranometer measurements, and changed the term 'brightness parameter' to 'clearness index'. Now, in Sect. 2.3 we introduce the clearness index, and in Sect. 3.4.1 we discuss its implementation in previous publications and ecosystem-atmosphere interactions related studies.

*3. The dataset used to develop (and test) the algorithm is quite limited. Actually, the authors already recognize this (lines 312-13 "This may be caused by the fact that the used total sky images were taken between 1 May and 31 July, leading to overrepresentation of summertime clouds. Thus, the number of undefined cases could increase in spring and autumn"). In addition, the authors decided not to use observations with SZA > 70 deg. I understand their concerns, but this threshold is usually set at 80 deg. In summary, admitting observations up to 80 deg SZA, and using more whole sky images, would allow a largest number of cases to be used in the algorithm development and validation.*

We have now further tested the springtime performance of the algorithm by taking additional 124 TSIs from March and April 2014 including also middle and high clouds (minimum CBH>2000 m). The method was similar as described in the Sect. 3.1: we visually classified the TSIs, and placed the transmittance
 and patchiness values of the cases in the plane of parameters (TR,PA). The results are shown below separately for low clouds, and middle and high clouds. We can see that the parameters mainly fall well to the parameter ranges displayed in Fig. 2 and Table 1. The results of the analysis are shown in the table.

[Figure]

[Figure]

| | | Algorithm | | | | | | | |
|---|---|---|---|---|---|---|---|---|---|
| | | Cu | St | Sc | Ns | Ac+As | Ci+Cc+Cs | Clear+Ci | Other | Agreement (%) |
| | Cu | 6 | 0 | 1 | 0 | 0 | 0 | 0 | 0 | 86 |
| | St | 1 | 13 | 1 | 0 | 0 | 0 | 0 | 2 | 76 |
| | Sc | 2 | 8 | 9 | 0 | 0 | 0 | 0 | 3 | 41 |
| Visual inspection | Ns | 0 | 0 | 0 | 0 | 0 | 0 | 0 | 0 | NaN |
| | Ac+As | 0 | 0 | 0 | 2 | 41 | 2 | 0 | 1 | 91 |
| | Ci+Cc+Cs | 0 | 0 | 0 | 0 | 0 | 4 | 4 | 7 | 27 |
| | Clear+Ci | 1 | 0 | 1 | 0 | 1 | 0 | 14 | 1 | 78 |
| | Total | | | | | | | | | 70 |

The discrepancies between St and Sc are mainly caused by cases when either both of them were present simultaneously, and it was hard to tell which one was more representative, or there was Ac or As layer on top of St/Sc clouds which affected the patchiness and transmittance readings. Similarly, the discrepancies between cirriform and clear sky cases from the introduced parameter ranges were mainly caused by partly cloudy conditions.

After this additional analysis, we are confident that the performance of the algorithm is good also during springtime, and therefore we removed the above-mentioned sentence considering the overrepresentation of the summertime cases. Furthermore, as the data availability of AERONET

data was low during March and April of 2016 and 2017, we produced median values of the data that we had, separately for spring (March+April) and September as explained in L. 135-139, we could define classes for many of the cases that had been in the "Base, no class" cloud class.

After careful consideration, we decided not to increase the SZA threshold from 70 deg to 80 deg as we could see that there was discrepancies between what the upward pointing ceilometer and the pyranometer measured at high SZAs. Cases falling into "No base, no class" cloud class 50 % had SZA > 60 deg, and in 25 % of the cases had SZA > 65 deg. Moreover, the further application of this algorithm is related to the studies investigating ecosystem-atmosphere interactions. SZA < 70 deg includes daytime data from March to September i.e. the whole growing season in southern Finland.

*Besides these three general comments, that should be addressed as comprehensively as possible, I do have a number of minor comments:*

*a. L. 14. "aerosol formation" or simply "aerosol load"*

We kept the original wording as we specifically were pointing to the possibility of clouds to affect aerosol formation as discussed in Dada et al. (2017), instead of more general concept of aerosol load/concentration in the presence of clouds.

*b. L. 53-57. This paragraph breaks the introduction. In my opinion, it could appear later, along with the content of the paragraph starting in L. 84.*

Referee is correct. We moved the paragraph as suggested.

*c. L. 84-89. In these 6 lines, the reference Duchon and O'Malley appears 4 times (!). I understand that this reference is important in the present study, but this is made clear by saying that the new algorithm is based on that previous work. You don't need to repeat it 4 times.*

We agree with Referee. We reduced the number of citations to Duchon and O'Malley (L. 84-90): "Our automatic cloud classification algorithm is based on global radiation and CBH measurements. It is an adaptation of the work by Duchon and O'Malley (1999). Their so called "pyranometer method", using only pyranometer data, was developed to classify clouds in places where no human observations were available. Even though the pyranometer method is simple and effective, its cloud type classes are rather broad (stratus, cumulus, cumulus+cirrus, cirrus, clear sky, precipitation+fog, and other), and the classification was found to be in agreement with human observations only 45 % of the time. Our improved cloud type classification algorithm uses additionally CBH data. Hence, the number of cloud type classes can be increased compared to Duchon and O'Malley (1999) because the clouds at different levels can be distinguished."

*d. L. 107, in the context of an applied Meteorology paper, the age of the pines is not relevant. Actually, it would be more relevant the height of trees and possible "shadows"on the instruments.*

Referee is correct. We removed the information of the age of the forest and added the information of canopy height as well as that the radiation measurements are conducted above the canopy level. The text now reads (L. 108):

"The station was established in 1995, and it is surrounded by Scots pine (Pinus sylvestris) dominated forest with canopy height of ca. 18 m (Hari and Kulmala, 2005).
The main data set in this work includes data from a pyranometer and a ceilometer. The pyranometer (Middleton solar SK08 pyranometer) measures global radiation at wavelengths of 0.3–4.8 µm. The measurements were conducted above the canopy level at SMEAR II."

*e. L. 118-119. You mention here that the Solis model is fed with AOD and precipitable water from Aeronet. This is ok, but then, this means that the method is not as easily "transferable" to other sites: they need to have not only pyranometer and ceilometer, but an Aeronet (Cimel sunphotometer) equipment too.*

This is a good notation, and we agree that the need of AOD and precipitable water for the clear sky model may complicate the transferability of the algorithm. It is possible, however, to change the clear sky model to some other that is better suitable for the environment or does not need extra variables. We added this information to the text (L. 118):
"We used Solis model because it explicitly takes into account the aerosol load in the atmosphere. However, in case AOD and precipitable water data are not available when applying this algorithm in other environments, other clear sky models may be employed, though we recommend to use as accurate model as possible."

*f. L. 124-125. Authors introduce here, too early, the idea of "parameter ranges". I think is not needed yet, before presenting the method for cloud classification.*

We agree with Referee, and changed the wording as follows (L. 127):
"In the development and validation process of the algorithm, we employed cloud classification made by human observer based on total sky images taken during the BAECC campaign between 01 May and 31 July 2014 (Fig. 1)."

*g. L. 141-142. I would say that the ceilometer is not at all (or hardly, if any) sensitive to SZA.*

Referee is correct. We changed the wording. The text now reads (L. 146):
"However, for the cloud occurrence and CBH analysis using the ceilometer measurements, we used data independent of the time of day and season, because unlike the pyranometer, the ceilometer is not sensitive to SZA."

*h. L. 156. Why not presenting and discussing here the need for "scaling" radiation measurements?*

We agree with Referee that leaving the discussion of the scaling for later complicated the readability unnecessary. The text was changed accordingly (L. 180).

*i. L. 157. I would start a new paragraph regarding TR_max. Otherwise, it might be confused with the CBH from ceilometer.*

Please, see answer to item j.

*j. L. 159. Well, before applying the classification you don't know if there are cumulus clouds. You should mention any type of varying cloudiness (obscuring and not obscuring the sun).*

After revising the algorithm, it was clear that TR_max was a redundant parameter, and hence we removed it completely, and thereby also the section discussing its use.

*k. L. 178. You could better use "irradiance on a horizontal surface" instead of the too generic "radiation".*

We changed the wording as suggested. The text now reads (L. 199):
"It is determined as a relation between the measured global irradiance and modeled irradiance on a horizontal surface at the top of the atmosphere:"

*l. Section 2. Is cloud occurrence from ceilometer data described in this methods section?*

It was not as the cloud occurrence from the ceilometer is simply the number of cases when the ceilometer detected a cloud base, but we now added clarifying sentence (L. 197):
"From the ceilometer data, the cloud occurrence is simply the number of cases when the ceilometer detected a cloud base."

*m. L. 200-201. Cloud occurrence from radiation data cannot show seasonality as 5 moths are missing.*

Referee is correct. We removed the sentence describing the variation in cloud occurrence from the pyranometer data.

*n. L. 224. How are middle and high clouds distinguished if there is a range (5000-7000m) where both are cloud types are included?*

We have now decreased the upper limit of middle clouds from 7000 m to 5000 m to avoid problems with distinguishing middle and high clouds. This had negligible effects on our results.

*o. L. 241. What do you mean by "uniformly and randomly"?*

We modified the sentence to make it easier to understand. The text now reads (L. 233):
"We took a sample of 665 total sky image–measurement data pairs randomly yet timewise uniformly, i.e. making sure that we utilized the whole measurement period, from among a set of total sky images taken between 01 May and 31 July 2014 in Hyytiälä."

*p. L. 244. "We first VISUALLY classified..."*

We changed the wording as suggested.

*q. Figure 4, caption. Explain the meaning of whiskers.*

We changed the caption to be more detailed:

"Illustration of the transmittance and patchiness ranges used as classification criteria for different cloud types. Markers display the locations of the maximum data point density of each cloud type, and whiskers extend to the lower and upper limits of the permitted parameter ranges, listed also in Table 1. Color shows the average CBH of each cloud type."

*r. L. 289. With the current form of Fig. 4, the U shape is difficult to catch.*

Referee is correct. We removed the paragraph.

*s. L. 341. "averaged over half an hour", but later you talk about 21 minutes. Please clarify (and consider my previous general comment #2)*

We thank Referee for pointing out the discrepancy in the text. The averaging was done over 21 min moving time window. We clarified the text and followed the general comment 2.

*t. Section 3.4, last paragraph. This is a somewhat confusing paragraph. If this paper is about cloud classification, why a discussion about ecosystem interactions? And, in this section the algorithm is not used, but the brightness parameter. Regarding the use of 0,7 as threshold, it is clearly too low to garantee a clear sky. In any case, if this is a discussion about results, it would fit better in section 3.5.*

The reason for discussing about ecosystem interactions comes from the idea of future application of the algorithm. However, we agree with Referee that the paragraph fits better to Sect. 3.5.

*u. L. 464-465. Well, this is a matter of which threshold you use. The "clearness index" in different versions is used to detect clear skies, by applying (if I'm right) a higher threshold (about 0,9).*

The selection of the suitable threshold is indeed depending on the application in question. Yet, the information of cloud variability is important as there can be situations when transmittance and clearness index are high e.g. in the presence of cumulus and cirriform clouds. We modified the text a bit to be more precise (L. 504):
"We found that cumulus, altocumulus, altostratus and cirriform clouds were present when clearness index was above 0.7 threshold that has been used as a limit for clear sky when studying aerosol–cloud interactions. Thus, the studies defining clear sky cases based on clearness index, may be biased. High clearness index threshold is deficient criterion as in the presence of patchy clouds, the clearness index may still be high periodically. Hence, the criterion should concern conditions with high transmittance and low patchiness."

Besides the comments by Referee, we changed term "transparency" to more generally used "transmittance". We changed the upper limit of transmittance for stratus clouds from 0.4 to 0.6. This was done because we could see that many St clouds fell into this area but were previously not classified. This, along with the applied median AOD and precipitable water values (see item 3), decreased the number of cases falling into the class "Base, no class". Now also the frequency of occurrence of stratus clouds is better in accordance with observations in *Climatic Atlas of Clouds Over Land and Ocean* (available online at https://atmos.uw.edu/CloudMap/, last access: 10 January 2020). We additionally changed the upper transmittance limit of Ns from 0.4 to 0.3 and lower

transmittance limit of Ac+As clouds from 0.4 to 0.3. This was done because we could see that especially in springtime Ac+As clouds were previously falsely classified as Ns. The overall occurrence of Ns clouds decreased (from 1.4 % to 0.6 %) and occurrence of Ac+As increased (from 9.0 % to 10.3 %) but otherwise the change did not affect our results.

[revised manuscript text omitted]